# MicroRNA 3′-compensatory pairing occurs through two binding modes, with affinity shaped by nucleotide identity and position

Sean E McGeary[1,2,3†], Namita Bisaria[1,2,3†], Thy M Pham[1,2,3], Peter Y Wang[1,2,3], David P Bartel[1,2,3]*

[1]Howard Hughes Medical Institute, Cambridge, United States; [2]Whitehead Institute for Biomedical Research, Cambridge, United States; [3]Department of Biology, Massachusetts Institute of Technology, Cambridge, United States

**Abstract** MicroRNAs (miRNAs), in association with Argonaute (AGO) proteins, direct repression by pairing to sites within mRNAs. Compared to pairing preferences of the miRNA seed region (nucleotides 2–8), preferences of the miRNA 3′ region are poorly understood, due to the sparsity of measured affinities for the many pairing possibilities. We used RNA bind-n-seq with purified AGO2– miRNA complexes to measure relative affinities of >1000 3′-pairing architectures for each miRNA. In some cases, optimal 3′ pairing increased affinity by >500 fold. Some miRNAs had two high-affinity 3′-pairing modes—one of which included additional nucleotides bridging seed and 3′ pairing to enable high-affinity pairing to miRNA nucleotide 11. The affinity of binding and the position of optimal pairing both tracked with the occurrence of G or oligo(G/C) nucleotides within the miRNA. These and other results advance understanding of miRNA targeting, providing insight into how optimal 3′ pairing is determined for each miRNA.

**\*For correspondence:**
dbartel@wi.mit.edu

[†]These authors contributed equally to this work

**Competing interest:** The authors declare that no competing interests exist.

## Editor's evaluation

This manuscript will be of interest to readers in the field of microRNA (miRNA) biology, particularly those interested in miRNA targeting. The authors interrogated non-canonical miRNA target recognition to a depth vastly exceeding any study to date. The results revealed unexpected, sequence-specific diversity in miRNA-targeting modes, providing new insights relevant for improved target prediction.

## Introduction

MicroRNAs (miRNAs) are ~22-nt regulatory RNAs that are processed from hairpin precursors. Upon processing, miRNAs associate with an Argonaute (AGO) protein and base-pair to sites within mRNAs to direct the destabilization and/or translational repression of these mRNA targets (*Jonas and Izaurralde, 2015*; *Bartel, 2018*). For most sites that confer repression in mammalian cells, pairing to miRNA nucleotides 2–7, referred to as the miRNA seed, is critical for target recognition, with an additional pair to miRNA position 8 or an A across from miRNA position 1 often enhancing targeting efficacy (*Lewis et al., 2005*; *Bartel, 2009*). Such sites with a perfect 6–8-nucleotide (nt) match to the miRNA seed region (*Figure 1A*, left) are heuristically predictive of repression, with longer sites being more effective than shorter ones and more sites being more effective than fewer sites (*Grimson et al., 2007*; *Agarwal et al., 2015*). In addition, contextual features extrinsic to a site itself can influence

**Figure 1.** Features of miRNA 3′-compensatory sites characterized using AGO-RBNS. (**A**) Pairing of typical canonical sites (left), 3′-supplementary, canonical sites (middle), and 3′-compensatory, noncanonical sites (right). Canonical sites contain contiguous complementarity (blue) to the seed (red). Sites with shifted complementarity (i.e., the 6mer-A1 and 6mer-m8 sites) are sometimes also classified as canonical sites. 3′-supplementary sites have pairing to the miRNA 3′ region, which supplements canonical seed pairing and is reported to be most effective if it centers on miRNA nucleotides 13–16 (green and orange). This 3′ pairing can supplement 8mer sites (as shown) as well as other canonical sites (not shown). 3′-compensatory sites resemble 3′-supplementary sites, except they lack perfect pairing to the seed and thus pairing to the 3′ region helps to compensate for this imperfect seed pairing. Vertical lines represent Watson–Crick pairing. (**B**) The architectures of 3′ sites. Three independent features define each architecture: (1) the length of 3′ pairing (left), measured as the number of contiguous base pairs to the miRNA 3′ region; (2) the position of 3′ pairing (middle-left), defined as the 5′-most miRNA nucleotide engaged in 3′ pairing; and (3) the offset between the seed pairing and 3′ pairing (middle), which specifies the number of unpaired nucleotides separating the seed- and 3′-paired segments in the target RNA relative to that in the miRNA. Mismatches to the seed pairing (middle-right) or within the 3′ pairing (right) can elaborate on these architectures, as can bulged nucleotides (not shown). (**C**) A programmed RNA library for using AGO-RBNS to examine 3′ pairing of let-7a. The library contains an 8-nt region with all 18 possible single-nucleotide mismatches (purple) to the let-7a seed (red), with 25 nt of random-sequence RNA upstream of this region and 5 nt of random-sequence RNA downstream. k-mer positions are numbered with respect to the programmed 8-nt mismatched site. B represents C, G, or U; D represents A, G, or U; V represents A, C, or G; N represents A, C, G, or U. The black vertical line depicts perfect pairing at position 8, and gray vertical lines indicate Watson–Crick matches at only five of the six seed positions. (**D**) The top 20 8-nt k-mers identified by AGO-RBNS performed with the highest concentration of AGO2–let-7a (840 pM) and the programmed library (100 nM). k-mers were ranked by the sum of their enrichments at the five positions of the library at which they were most enriched. Left, alignment of k-mers, indicating in pink nucleotides that were not Watson–Crick matches to the miRNA. Right, heat map showing k-mer enrichment at each position of the library, with pairing shown for the top 8-nt k-mer at the position of its greatest enrichment. Black vertical lines depict perfect Watson–Crick pairing, and gray vertical lines indicate Watson–Crick matches at only five of the six seed positions.

targeting efficacy (*Brown et al., 2005*; *Ameres et al., 2007*; *Grimson et al., 2007*; *Kedde et al., 2007*; *Nielsen et al., 2007*; *Saetrom et al., 2007*; *Tafer et al., 2008*; *Kedde et al., 2010*; *Wan et al., 2014*; *Agarwal et al., 2015*; *McGeary et al., 2019*).

Pairing to the miRNA 3′ region, particularly pairing that includes miRNA nucleotides 13–16, can supplement perfect seed pairing to enhance targeting efficacy beyond that of seed pairing alone, and extensive pairing to the 3′ region can compensate for imperfect seed pairing to enable consequential repression (*Brennecke et al., 2005*; *Lewis et al., 2005*; *Grimson et al., 2007*). These two bipartite site types are referred to as 3′-supplementary and 3′-compensatory sites, respectively (*Figure 1A*, middle and right). Although 3′-supplementary sites are less common than sites with only a seed match, comprising ~5% of all conserved sites observed in mammals, thousands of sites with preferentially conserved 3′-supplementary pairing are present in human 3′ UTRs (*Grimson et al., 2007*; *Friedman et al., 2009*). Conserved 3′-compensatory sites are even less common, comprising only ~1.5% of all preferentially conserved sites observed in human 3′ UTRs (*Friedman et al., 2009*). Nonetheless, two instances of this relatively rare site type within the 3′ UTR of *lin-41* mediate the extreme morphological and developmental defects by which the *let-7* miRNA was discovered in *C. elegans* (*Pasquinelli et al., 2000*; *Reinhart et al., 2000*; *Ecsedi et al., 2015*). Moreover, the use of these 3′-compensatory sites rather than canonical sites for *lin-41* repression is consequential; site mutations that create perfect seed pairing while maintaining the 3′ pairing cause precocious repression of the mRNA by other members of the let-7 seed family expressed during earlier larval stages (*Brancati and Großhans, 2018*). These results support the notion that 3′-compensatory sites enable differential target specificity between miRNAs that share a common seed sequence but differ within their 3′ regions (*Brennecke et al., 2005*; *Lewis et al., 2005*).

Although global analyses of site conservation and efficacy provide compelling evidence that pairing to the miRNA 3′ region is also utilized in mammalian cells (*Friedman et al., 2009*), these approaches have limitations for evaluating which 3′-pairing architectures are most effective, due to the vast number of 3′-pairing architectures that are possible for a single miRNA sequence. The pairing architecture of a 3′-compensatory site can be described by five characteristics: (1) the length of contiguous pairing between the site and the miRNA 3′ region, (2) the position of pairing to the miRNA 3′ region, as defined by the 5′-most miRNA nucleotide involved in 3′ pairing, (3) the difference between the number of unpaired target nucleotides and the number of unpaired miRNA nucleotides bridging the seed and 3′ pairing, hereafter referred to as the '3′-pairing offset,' (4) the nature of the imperfect pairing to the seed, and (5) the nature of any imperfections in the 3′ pairing (*Figure 1B*). When considering only sites with perfect 3′ pairing with lengths ranging from 4 to 11 base pairs (bp) at all possible 3′ positions, offsets ranging from −4 to +16 nt, and seed pairing interrupted by one of 18 possible single mismatches (or wobbles) to the 6-nt seed, there are >16,000 possible variants to the site architecture. However, for each miRNA, most of these possibilities are not present even once in all the 3′ UTRs of a transcriptome. Thus, data from multiple miRNAs must be aggregated to observe a reliable signal of either efficacy or conservation, which prevents identification of miRNA-specific pairing preferences. Indeed, even when aggregating multiple miRNA-perturbation (e.g., transfection) datasets, which enables efficacy of 3′-supplementary sites to be detected (*Grimson et al., 2007*), a signal for the efficacy of 3′-compensatory sites has not been reported, underscoring the challenge of using global analyses of conservation or repression efficacy to determine which architectures are more effective than others.

The observation that miRNA targeting efficacy observed in the cell is largely a function of the affinity between AGO–miRNA complexes and their sites (*McGeary et al., 2019*) indicates that contributions of 3′ pairing to affinities measured in vitro can provide insight into biological targeting efficacy. Early measurements showed that pairing to positions 13–16 of let-7a imparts only a twofold increase in binding affinity, which led to the view that 3′-supplemental pairing contributes only modestly to affinity (*Wee et al., 2012*). Further measurements revealed some differences between miRNAs, with the observation that pairing to positions 13–16 of miR-21 increases affinity by 11-fold (*Salomon et al., 2015*), and a striking effect of longer pairing, with the observations that 10 bp of 3′-supplementary pairing to miR-122 and 9 bp of 3′-supplementary pairing (including a terminal G:U wobble) to miR-27a increases affinity by 20- and >400-fold, respectively (*Sheu-Gruttadauria et al., 2019a*). Other measurements illustrate the influence of the length of the target segment bridging the seed and 3′ pairing, with binding affinity varying ~10-fold as this length is varied over a range of 1–15

nt (*Sheu-Gruttadauria et al., 2019b*). Taken together, these reports demonstrate the potential for miRNA 3′ pairing to enable high-affinity binding, and also illustrate that the benefit of this pairing varies, depending on the miRNA sequence and 3′-pairing architecture. Understanding how these features together modulate the benefit of 3′ pairing will be possible only after acquiring many more measurements with multiple miRNA sequences.

Imaging-based, high-throughput single-molecule biochemistry has recently been applied to acquire affinity measurements for ~23,000 sites for each of two miRNAs (let-7a and miR-21), including many sites with 3′ pairing (*Becker et al., 2019*). These measurements revealed that miR-21 relies more on 3′ pairing when binding to a fully complementary target than does let-7a, that homopolymeric insertions are the least disruptive to binding when inserted between nucleotides 8 and 11 within the context of fully complementary binding, and that mismatches near the miRNA 3′ terminus (after position 16) decrease binding affinity but increase target slicing. However, because the design of target libraries was based primarily on fully complementary RNA targets to which varying extents of mismatched, bulged, and deleted nucleotides were introduced, only a small minority of the possible 3′-pairing architectures were queried.

RNA bind-n-seq (RBNS) enables unbiased, high-throughput assessment of binding sites embedded within a larger random-sequence context (*Lambert et al., 2014*; *Dominguez et al., 2018*). We recently adapted RBNS for the study of miRNA targeting, and we built an analysis pipeline enabling calculation of relative equilibrium dissociation constants ($K_D$ values) for many thousands of different RNA $k$-mers ≤12 nt in length (*McGeary et al., 2019*). Here, we further adapted the AGO-RBNS protocol to enable examination of sites >12 nt in length, thereby enabling the high-throughput investigation of bipartite sites containing near-perfect seed pairing and 4–11 additional pairs to the miRNA 3′ region. We applied this modified protocol to the systematic interrogation of the contribution of 3′ pairing for three natural miRNA sequences and four synthetic derivatives. We also performed a massively parallel reporter assay, which confirmed that key observations derived from affinities measured in vitro apply also to repression in cells.

## Results

### RBNS measures affinities for many 3′-compensatory sites of let-7a

AGO-RBNS begins with a series of 4–6 binding reactions, each containing an RNA library at a fixed concentration and a purified AGO–miRNA complex at a variable concentration spanning a 100-fold range (*McGeary et al., 2019*). Each molecule of the RNA library has a central region of random-sequence nucleotides flanked by constant sequences on each side that enable preparation of sequencing libraries. Upon reaching binding equilibrium, each reaction is passed through a nitrocellulose membrane, which retains AGO–miRNA complexes and any library molecules that are bound to the complexes. These bound library molecules are isolated and subjected to high-throughput sequencing, along with the input RNA library. Binding of an individual $k$-mer can be detected as enrichment in the bound compared to input sequences, and relative $K_D$ values can be estimated simultaneously for hundreds of thousands of different $k$-mers by fitting a biochemical model to $k$-mer fractional abundances from each of the bound libraries.

As originally implemented, AGO-RBNS cannot provide reliable information on sites with more than ~5 supplementary/compensatory pairs because such sites, which involve >12 bp of total pairing (*Figure 1A*, middle and right), are too rare in the sequences obtained from the input RNA library to enable accurate calculation of enrichment values. To overcome this constraint for sites to let-7a, a miRNA with physiologically relevant 3′ pairing (*Pasquinelli et al., 2000*; *Reinhart et al., 2000*; *Brancati and Großhans, 2018*), we used a library that contained a programmed region of imperfect seed pairing to let-7a, with 25 and 5 nt of random-sequence RNA separating the programmed region from the 5′ and 3′ constant sequences, respectively (*Figure 1C*). In each library molecule, this programmed region of imperfect seed pairing contained a let-7a 8mer site with a mismatch at one of its six seed nucleotides, such that each library molecule had one of 18 possible single-nucleotide seed mismatches (including wobbles) in approximately equal proportion. With this programmed region of imperfect seed pairing, each library contained 3′-compensatory sites at an ~250-fold greater frequency than expected for a fully randomized RNA library.

AGO-RBNS was performed using this programmed library and purified AGO2–let-7a, choosing AGO2 from among the four human AGO paralogs because of its relatively high expression (***Völler et al., 2016***; ***Müller et al., 2019***) and for comparison to previous biochemical and structural studies that use human or mouse AGO2 (***Schirle and MacRae, 2012***; ***Wee et al., 2012***; ***Schirle et al., 2014***; ***Schirle et al., 2015***; ***Chandradoss et al., 2015***; ***Salomon et al., 2015***; ***Klum et al., 2018***; ***Becker et al., 2019***; ***McGeary et al., 2019***; ***Sheu-Gruttadauria et al., 2019a***; ***Sheu-Gruttadauria et al., 2019b***). For our initial analysis, we calculated the enrichment of all 8-nt *k*-mers at each position between the programmed region and the 5′-constant region of the library, after first removing reads with any of the six canonical sites to let-7a. The enriched *k*-mers had substantial complementarity to the 3′ region of let-7a (***Figure 1D***). The most enriched was AUACAACC—the perfect Watson–Crick match to positions 11–18 of let-7a (***Figure 1D***). This 8-nt 3′ site was most strongly enriched when starting at position 15 of the library, which suggested that an internal loop with two miRNA nucleotides (9 and 10) and six target-site nucleotides (positions 9–14) separating seed pairing and 3′ pairing was optimal (***Figure 1D***, top). Using our nomenclature (***Figure 1B***), this 3′ site was classified as a position-11 site with pairing length of 8 bp and offset of +4 nt. Note that here and throughout this study we refer to contiguous complementarity as 'pairing,' even though constraints imposed by AGO2 might prevent physical pairing from occurring at some complementary positions. This 8-nt, position-11 site was also ≥5-fold enriched at seven other neighboring offsets (corresponding to library positions 8–15), indicating that looping out 3–10 unpaired library nucleotides opposite miRNA nucleotides 9 and 10 was tolerated, albeit to varying degrees (***Figure 1D***).

The second-most enriched 8-nt *k*-mer was UACAACCU—the perfect Watson–Crick match to let-7a positions 10–17 (***Figure 1D***). This 3′ site had a maximal enrichment with five, rather than six, unpaired library nucleotides spanning the seed and 3′ pairing, with the distribution of enrichments shifted by 1 nt in comparison to that of the AUACAACC site. This 1-nt shift in the enrichment distribution corresponded with the 1-nt shift in site position (from 11 to 10 of the miRNA) to maintain an offset of +4 target nucleotides. Indeed, the next 18 most enriched 8-nt *k*-mers represented 3′ sites with pairing positions ranging from miRNA nucleotides 9–12, with enrichment distributions that correspondingly shifted to reflect an overall optimal offset of +4 target nucleotides (***Figure 1D***). Each had a contiguous stretch of 6–8 perfect Watson–Crick pairs to the let-7a 3′ region, usually including the ACAACC *k*-mer, which suggested that perfect pairing to let-7a positions 11–16, with a +4 nt offset, was particularly effective for enhancing site affinity.

## let-7a has two distinct 3′-pairing modes

For a more comprehensive examination of 3′ sites of varied lengths, positions, and offsets (***Figure 1B***), we enumerated 3′ sites of lengths 4–11 nt that perfectly paired to the miRNA starting at any position downstream of nucleotide 8. For each length and position of 3′ pairing (e.g., for the 8mer-m11–18), we further enumerated all pairing offsets compatible with the 3′ site residing within the 25-nt random-sequence region upstream of the programmed site, converting each library position to an offset value based on the pairing position of each 3′ site (***Figure 2A***). For our initial $K_D$ estimation and analyses, we pooled the reads for the 18 possible seed-mismatch types. This pooling increased the read counts for each 3′-pairing architecture, which enabled examination of sites as long as 11 nt, which in turn enabled analysis of 1006 distinct 3′-pairing architectures.

Simultaneous estimation of the fractional abundance of these sites in each of the AGO2–let-7a-bound libraries in comparison to that of the input library enabled calculation of their relative $K_D$ values. As illustrated for the 8-nt *k*-mer identified as most enriched in the previous analysis (***Figure 1D***, top row), variation in $K_D$ values qualitatively tracked with that of enrichment values but quantitatively differed due to the attenuating effects of background binding and site saturation on enrichment values (***McGeary et al., 2019***; ***Figure 2A***). Relative $K_D$ values corresponding to a broad spectrum of 3′-pairing architectures spanned a several hundred–fold range, with strong agreement observed between the results of replicate experiments performed independently with different preparations of both AGO2–let-7a and the let-7a programmed library ($r^2$ = 0.96, n = 1477; ***Figure 2—figure supplement 1A***, left). Agreement between the two replicates was maintained, albeit to a lesser degree, when read counts for each 3′-pairing architecture were not pooled over the 18 seed-mismatched sites in the programmed region ($r^2$ = 0.78, n = 23,912; ***Figure 2—figure supplement 1A***, right). Furthermore, for shorter 3′ sites, which could be analyzed using data from a standard AGO-RBNS experiment

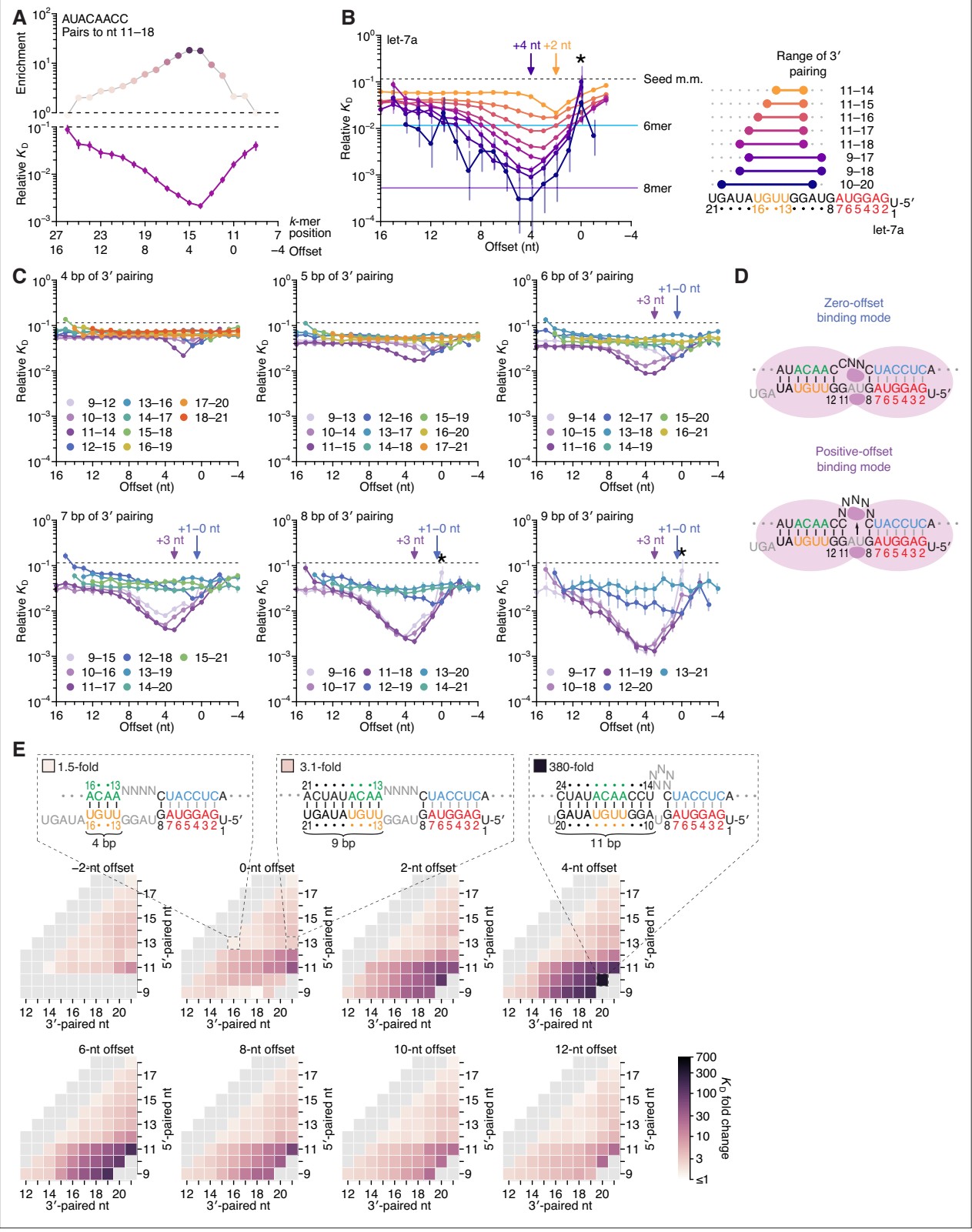

**Figure 2.** Pairing to nucleotide 11 and a positive offset promote high-affinity binding to let-7a.in P. (**A**) Correspondence of enrichment and relative $K_D$ value of sites with the AUACAACC $k$-mer (the perfect match to miRNA positions 11–18) measured at each position in the programmed library. Each of these positions (upper $x$-axis) corresponds to the indicated offset (lower $x$-axis). For example, because this $k$-mer paired to miRNA positions 11–18, pairing beginning at $k$-mer position 11 had a 0-nt offset. The $k$-mer enrichments and their associated colors (top) correspond to those of the top row

*Figure 2 continued on next page*

*Figure 2 continued*

of *Figure 1D*. For details on how relative $K_D$ values were calculated for each site possibility, see *McGeary et al., 2019*, Figure 1C–E and Materials and methods section 11. (**B**) Relative $K_D$ values of let-7a 3′-compensatory sites that had optimally positioned 3′ pairing of 4 (orange) to 11 (dark blue) bp. For each length of 3′ pairing, the optimal position is shown in terms of its complementarity to let-7a (right). For each of the 3′-compensatory sites, the relative $K_D$ value is plotted as a function of its offset (left), as done for sites with 8 bp of optimally positioned 3′ pairing in (**A**). Vertical lines indicate 95% confidence intervals. The dashed horizontal line indicates the geometric mean of the 18 relative $K_D$ values of the seed mismatch sites, each calculated from reads with <4 nt of contiguous complementarity to the miRNA 3′ region. The horizontal blue and purple lines indicate the relative $K_D$ values of the canonical 6mer and 8mer sites, respectively. The arrows at +2 and +4 nt mark a shift in the optimal offset observed with increasing 3′-pairing length. The asterisk denotes the anomalously low binding affinity measured for 3′ sites that pair contiguously with seed pairing (i.e., sites with pairing at position 9 with an offset of 0 nt). (**C**) The dependency of let-7a 3′-pairing affinity on pairing length, position, and offset. Each panel shows the relative $K_D$ values of 3′-compensatory sites with 3′ pairing of a specified length over a range of positions and offsets. Each trend line is colored according to pairing position, spanning positions 9 (light violet) to 18 (red) when possible. The arrows between 0 and 1 nt and at +3 nt mark a shift in the optimal offset as the position of 3′ pairing shifted to include nucleotide 11 of let-7a. Otherwise, these panels are as in (**B**, left). (**D**) Schematics of the two 3′-binding modes. In the zero-offset binding mode (top), miRNA nucleotide 11 is inaccessible due to occlusion by the central region of the AGO protein. In the positive-offset binding mode (bottom), the longer stretch of bridging target nucleotides enables a conformation in which nucleotide 11 is available for pairing to the target RNA. Although not intended to accurately reflect the conformation of either binding mode, these schematics illustrate how a larger offset might enable pairing to a more centrally located miRNA nucleotide. (**E**) Affinity profile of the let-7a 3′ region. Each cell indicates the fold change in relative $K_D$ attributed to a 3′ site with indicated length, position, and offset of pairing. Each row within a heat map corresponds to a different miRNA nucleotide at the start of the 3′ pairing, and each column corresponds to a different miRNA nucleotide at the end of the 3′ pairing. Each heat map shows the results for a different offset. The three diagrams indicate the fold-change values and architectures for 3′ sites pairing to miRNA nucleotides 13–16 with an offset of 0 nt (left), pairing to miRNA nucleotides 13–21 with an offset of 0 nt (middle), and pairing to miRNA nucleotides 10–20 with an offset of +4 nt (right). Gray boxes indicate pairing ranges that were either too short (<4 bp) or too long (>11 bp) for relative $K_D$ values to be reliably calculated. Black vertical lines depict perfect Watson–Crick pairing, and gray vertical lines indicate Watson–Crick matches at only five of the six seed positions.

The online version of this article includes the following figure supplement(s) for figure 2:

**Figure supplement 1.** Reproducibility of AGO-RBNS with programmed libraries and correspondence with random-sequence libraries.

**Figure supplement 2.** Re-analysis of binding experiments performed in *Becker et al., 2019*.

that used a non-programmed random-sequence library (*McGeary et al., 2019*), the relative $K_D$ values determined from the programmed library correlated well with those determined from a random-sequence library ($r^2 = 0.83$, *Figure 2—figure supplement 1B*). Despite the overall correlation, a minor systematic difference in the values for the same sites determined from the two types of libraries was observed. This distortion was presumed to be due to the absence of library RNA molecules containing no site and was corrected accordingly (*Figure 2—figure supplement 1B*).

To investigate the interplay of pairing position, length, and offset, we identified the optimal 3′ sites of lengths 4–11 nt and, as in *Figure 2A*, examined the effect of varying offset on the affinity of each of these sites (*Figure 2B*). Nearly all possibilities examined had values readily distinguished from the log-averaged value for seed-mismatched sites alone, with compensatory pairing to miRNA nucleotides 11–16 at optimal offsets yielding binding affinities comparable to that of the canonical 6mer (*Figure 2B*, left). Further inspection of longer 3′ sites underscored the conclusion that pairing to the GGUUGU segment spanning positions 11–16 of let-7a is the most consequential for 3′-compensatory pairing, as all optimal pairing positions for 3′ sites ≥6 nt in length paired to this segment. Moreover, inspection of the optimal positions for shorter sites showed that pairing to the 5′ end of this segment (containing the sequence GGUU) was more impactful than pairing to its 3′ end (*Figure 2B*, right). In addition, increasing the length of pairing from 4 to 11 bp led not only to increased binding affinity at almost all offsets, as might have been expected, but also to a shift in the optimal offset, with a preferred offset of +2 nt when pairing with 4 bp compared to a preferred offset of +4 nt when pairing with 9–11 bp (*Figure 2B*, left).

To investigate further the interplay between affinity, pairing position, and pairing offset, we plotted the relative affinities of all possible positions and offsets for let-7a 3′ pairing of lengths ranging from 4 to 9 bp (*Figure 2C*). These plots revealed a striking change in the affinities and preferred offsets as pairing shifted from position 12 to position 11. For 3′ sites of each length, those that began at let-7a position 12 (dark blue points of *Figure 2C*) had intermediate affinity and optimal offsets of 0 or +1 nt, with clearly reduced affinity as offsets increased beyond +1 nt. At these offsets of 0 or +1 nt, 3′ sites that began at position 11 (dark purple points) had affinities similar to those that began at position 12. However, in stark contrast to the sites beginning at position 12, sites beginning at 11 had strikingly increased affinity at more positive offsets, with affinity peaking at offsets of +2 or +3 nt. Sites

beginning at +10 were similar, with affinity peaking at an offset of +4 nt. These results suggested that pairing to position 11 in the central region of the miRNA is less accessible than pairing to position 12, and therefore a longer loop in the target sequence is required to bridge seed pairing with 3′ pairing that includes position 11 (*Figure 2D*). Nonetheless, when the increased offset enables pairing to position 11, substantially greater affinity can be achieved. We call this newly defined binding mode, which includes pairing to miRNA position 11 and greatly benefits from short positive offsets, the 'positive-offset' binding mode. Accordingly, the more conventional binding mode, which lacks pairing to position 11 and does not benefit from offsets greater than +1 nt, we call the 'zero-offset' binding mode.

Some of the weakest relative affinities were observed for extended 3′-pairing possibilities that began at position 9 with an offset of 0 nt (*Figure 2B and C*, asterisks). These weak values were attributable to AGO2-catalzyed slicing of molecules with extensive contiguous pairing, which would have depleted these molecules from our bound library. Supporting this idea, analogous sites with offsets of either −1 or +1 nt, which were expected to disrupt slicing due to single-nucleotide bulges in either the miRNA or the site, respectively, did not have aberrantly low relative affinities. This idea was also consistent with reports that AGO2 can slice sites that have a seed mismatch but are otherwise extensively paired to the guide RNA (*Wee et al., 2012*; *Chen et al., 2017*; *Becker et al., 2019*).

We next used heat maps to visualize the interplay between 3′-site position and pairing length at different offsets (*Figure 2E*). Within each heat map, a difference between adjacent cells corresponded to the difference in $K_D$ fold change caused by the addition or removal of a pair at either the 5′ end (adjacent rows) or the 3′ end (adjacent columns) of the 3′ site, while maintaining the same offset. For example, in the heatmaps corresponding to offsets of +2 to +12 nt, the prominent contrast between the row corresponding to pairing beginning with nucleotide 11 and the row corresponding to pairing beginning with nucleotide 12 illustrated the strong benefit of pairing to G11 of let-7a (*Figure 2E*). At the optimal offset length of +4 nt, pairing to let-7a positions 10–20 conferred an ~380-fold increase in affinity over the average seed-mismatched site alone (*Figure 2E*), leading to an overall binding affinity rivaling that of the canonical 8mer (*Figure 2B*). The binding affinity of this site and all other sites decreased nearly uniformly as offset values increased beyond +4 nt. Binding affinity decreases were less uniform as offset values decreased to 0 and –2 nt, which reflected a switch from the positive-offset binding mode to the zero-offset binding mode, with a concomitant reduction in the benefit of pairing to nucleotide 11.

Previous low-throughput measurements of the benefit of 3′ pairing for let-7a examined the influence of pairing to miRNA positions 13–16 at an offset of 0 nt and found that this pairing confers a 1.6–2-fold increase in binding affinity (*Wee et al., 2012*; *Salomon et al., 2015*). Likewise, our measurements for this 4-nt 3′ site indicated that it conferred a 1.5-fold increase in affinity (*Figure 2E*). Furthermore, maintaining the offset of 0 nt and the pairing position of 13 and extending pairing to the very 3′ end of let-7a increased the binding affinity to only 3.1-fold (*Figure 2E*). These results highlight the importance of both a positive offset and pairing to position 11 of let-7a—two features that would have been difficult to identify without comprehensive investigation of the 3′-pairing preferences of this miRNA. Indeed, the importance of these two features is not revealed in an analysis of a dataset that reports the affinities of ~23,000 different sites to let-7a, because these ~23,000 sites were not designed to analyze the combined effects of varying both pairing position and pairing offset (*Becker et al., 2019*; *Figure 2—figure supplement 2*).

## Pairing preferences of let-7a correspond with repression efficacy in cells

We next tested whether features associated with higher affinity also conferred greater repression in cells. Our analysis centered on 15 different 3′-compensatory sites, designed to test the consequences of changing the position, length, and/or offset of 3′ pairing (*Figure 3A*). These sites were each placed in the 3′ UTR of a reporter, at either an upstream position, a downstream position, or at both the upstream and the downstream positions (*Figure 3A*). For comparison, we analyzed five sites with only seed pairing and seven no-site sequences that had no more than five contiguous pairs to let-7 (*Figure 3A*). We also analyzed the dual 3′-compensatory sites that mediate *lin-41* repression in *C. elegans* (*Figure 3A*; *Reinhart et al., 2000*). To account for the effects of local sequence context, sites were each placed within 14 different sequence contexts, one of which was the native sequence context of sites in the 3′ UTR of *C. elegans lin-41* (*Figure 3A*).

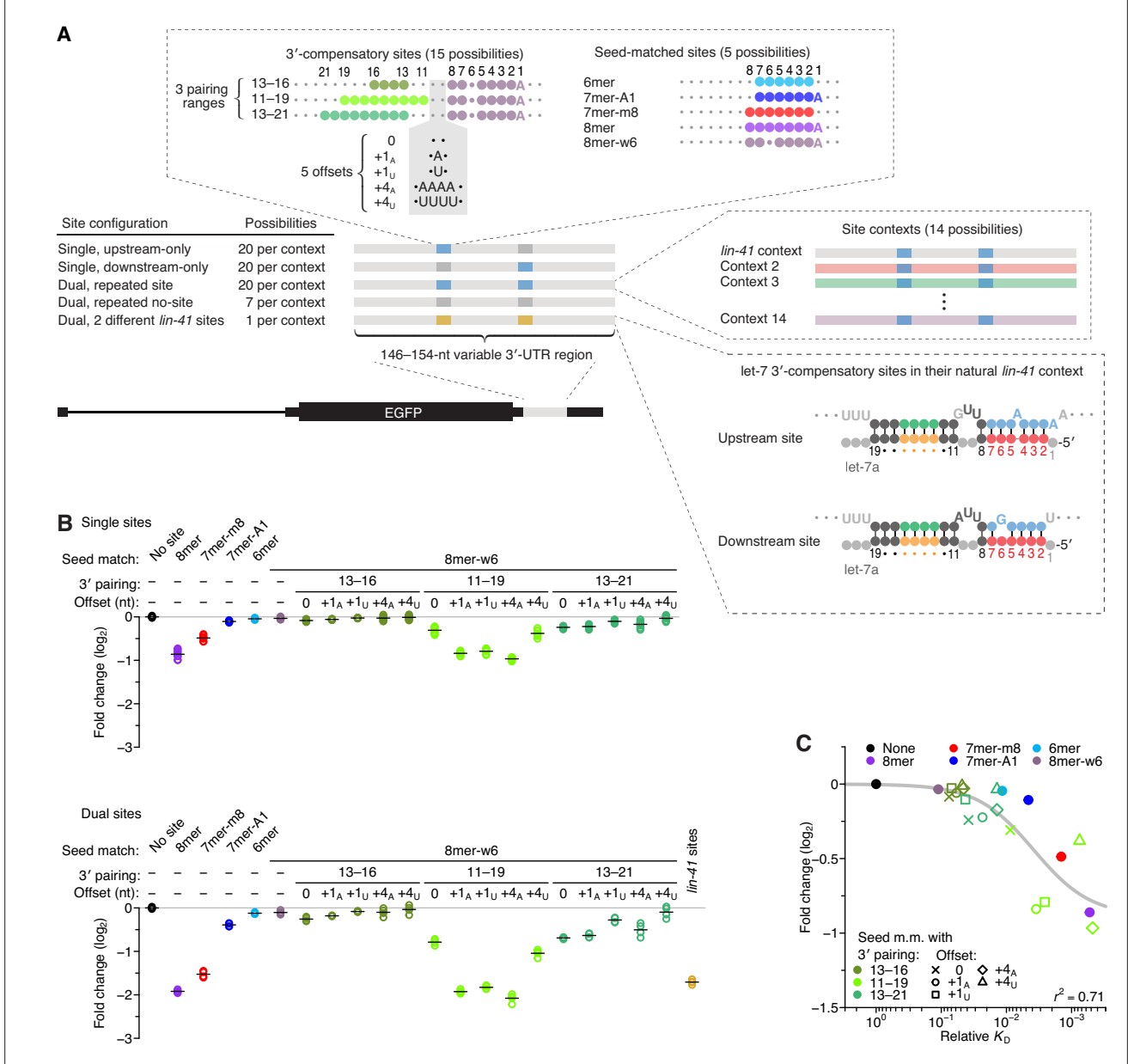

**Figure 3.** Interplay between the effects of length, position, and offset of 3′ pairing, as measured for let-7a by comparing efficacy of repression in cells. (**A**) Design of reporter mRNAs. For the diagrams of 3′-compensatory sites and seed-matched sites, a large colored circle indicates a Watson–Crick match to let-7a, a smaller plum circle indicates a G wobble across from U at position 6 of let-7a, and a small gray circle indicates lack of complementarity. The diagram for the *lin-41* sites is as in *Figure 1A*. Gray nucleotides and small gray circles indicate positions allowed to vary in the 14 different contexts. Each of the nonzero offset possibilities (i.e. the +1A, +1U, +4A, and +4U) was formed by inserting the indicated nucleotides between the two nucleotides opposite those of let-7a positions 9 and 10. (**B**) Repression attributed to each site type after co-transfecting let-7a into F9 cells. F9 cells were chosen for this experiment because they endogenously express very little let-7 (*Mayr et al., 2007*). Changes observed upon let-7a co-transfection are plotted for reporters with the single-site configurations (top) and for those with the dual-site configuration (bottom). Mean values are represented by horizontal black lines. For the single-site analysis, changes associated with upstream-only and downstream-only site configurations were plotted separately to yield eight values spanning the four replicate experiments. Changes were normalized to the mean no-site value. (**C**) The relationship between repression observed in cells and relative $K_D$ values derived from AGO-RBNS. The line represents a fitted model relating binding affinity to predicted repression ($r^2$, coefficient of determination of the model fit).

The online version of this article includes the following figure supplement(s) for figure 3:

**Figure supplement 1.** Negligible repression observed when co-transfecting a control miRNA, miR-1.

**Figure supplement 2.** Enhanced repression observed for each site architecture when presented within the native *lin-41* 3′-UTR context.

Plasmids designed to express these 952 different reporter variants were co-transfected into F9 cells with either a let-7a duplex, a control miR-1 duplex, or no miRNA duplex (the mock co-transfection), and accumulation of each variant in the presence of each co-transfected miRNA was monitored by high-throughput sequencing and compared to accumulation observed in the mock co-transfection (*Figure 3B* and *Figure 3—figure supplement 1*). Most of the conclusions regarding binding affinities inferred from AGO-RBNS also held with respect to repression in cells, including the marginal benefit of pairing to only nucleotides 13–16 of let-7a, the greater benefit of pairing to let-7a nucleotides 11–19 compared to nucleotides 13–21, the strong benefit of a positive offset when pairing to nucleotides 11–19 of let-7a but not when pairing to nucleotides 13–16 or 13–21, and the ability of extended 3'-compensatory pairing at a favorable position and offset to impart activity matching that of the canonical 8mer site (*Figure 3B*).

Among the sites with pairing to nucleotides 11–19 of let-7a, the most effective was the one with an offset of +4 nt formed by insertion of four consecutive A nucleotides within the segment of the target that linked seed and 3' pairing (*Figure 3B*, $+4_A$). This site was more effective than the two possibilities with an offset of +1 formed by insertion of either a single A or U nucleotide ($+1_A$ and $+1_U$, respectively; $p < 0.02$, Tukey's range test), which were in turn more effective than the site that lacked an insertion (0) and thus had a 0-nt offset (*Figure 3B*, $p < 10^{-4}$, Tukey's range test). Although this rank ordering was consistent with that predicted from relative affinities, some quantitative disagreement with the relative affinities was observed, with the three sites with positive offsets performing substantially better than expected from their relative affinities (*Figure 2C*). Another notable divergence from the binding results was the poorer-than-expected efficacy of the site with a +4 nt offset formed by the insertion four consecutive U nucleotides ($+4_U$). Efficacy of this site was less than half of that of the other sites with positive offsets, and its greater efficacy over the site with a 0-nt offset was statistically significant only for the dual-site configuration ($p < 10^{-4}$, Tukey's range test). For the 3'-compensatory site with 9 nucleotides of complementarity starting at position 13, the efficacy of the $+4_U$ variant was also less than that of the $+4_A$ variant (*Figure 3B*, $p < 0.01$, Tukey's range test). These results indicated that the primary nucleotide identity of the segment that links seed and 3' pairing can modulate repression. One way this modulation might occur is through the action of RNA-binding proteins, many of which prefer short oligo(U) tracts (*Dominguez et al., 2018*; *Van Nostrand et al., 2020*), as binding of a protein to this segment would be expected to interfere with 3' pairing.

Compared to the single-site configurations, the dual-site configuration yielded greater repression, with an average increase of ~2.8-fold (*Figure 3B*), implying some cooperativity in the action of the two sites (*Grimson et al., 2007*; *Saetrom et al., 2007*; *Broderick et al., 2011*; *Briskin et al., 2020*). The most effective synthetic 3'-compensatory sites tested (those with complementarity to nucleotides 11–19 and with a $+1_A$-nt or $+4_A$-nt offset) were more repressive on average than the two sites found within *lin-41* mRNA of *C. elegans* ($p < 10^{-3}$, 1.2-fold more repressive). The *lin-41* sites, as well as the other effective sites, were all more effective when examined in the *lin-41* local sequence context than they were in most of the other contexts (*Figure 3—figure supplement 2*).

Overall, we found that affinity observed in vitro corresponded well to repression observed in cells (*Figure 3C*, $r^2 = 0.71$). This correspondence for 3' sites resembled that observed for seed-matched sites (*McGeary et al., 2019*) and provided counterevidence to a recent proposal that 3' pairing might be preferentially destabilized in cells (*Bibel et al., 2022*). When framed in the context of the 3'-compensatory sites acting in the *C. elegans lin-41* mRNA, our results indicate that the developing animal exploits 3'-compensatory pairing at a favorable position (position 11) of let-7, which is enhanced through the positive-offset binding mode to confer robust repression of the *lin-41* mRNA, with repression further enhanced by favorable site context and some inter-site cooperativity.

## Different miRNAs have distinct 3'-pairing preferences

The optimal 3'-pairing architecture for let-7a differed from that previously elucidated for miRNAs more generally (*Grimson et al., 2007*). When pooling repression and conservation data for 11 miRNAs, pairing to miRNA nucleotides 13–16, with an offset of 0 nt appears to be most consequential (*Figure 1A*; *Grimson et al., 2007*). Because the previous analysis represents the average of trends derived from multiple miRNAs, a diversity of miRNA-specific 3'-pairing preferences might explain this disagreement. We therefore measured the 3'-pairing profiles of two other well-studied miRNAs, miR-1 and miR-155, for comparison with the let-7a profile.

Stabilizing 3′ pairing was observed for both miR-1 (*Figure 4A*) and miR-155 (*Figure 4B*), with binding affinity increasing with the length of pairing, as observed for let-7a (*Figure 2*). However, the magnitude of increased binding affinity differed from that of let-7a and that of each other: the affinity of 3′ pairing to miR-1 was more modest, with 3′-compensatory sites seldomly reaching the affinity of its canonical 6mer site (*Figure 4A*), whereas for miR-155, they often reached the affinity of its canonical 8mer site, and in some cases increased affinity by >500-fold (*Figure 4B*). The positions of the best sites at each length also differed from those of let-7a. For miR-1, optimal 4-nt sites paired to miRNA nucleotides 12–15, and when considering optimal sites of increasing lengths, pairing extended continuously, primarily toward the 3′ end of the miRNA and never reaching miRNA nucleotide 10 (*Figure 4A*, right). By contrast, for miR-155, optimal 4-nt sites paired to miRNA nucleotides 13–16, and for optimal sites of increasing lengths, pairing sometimes shifted discontinuously and never included miRNA nucleotide 12 (*Figure 4B*, right).

Analysis of each of the optimal 3′ sites of miR-1 and miR-155 along the length of the random region indicated that, unlike sites for let-7a, those for neither of these two miRNAs underwent a significant shift in the preferred offset (*Figure 4A and B*, left). Nevertheless, the longer optimal sites of miR-1 extended to position 11, and their range of near-optimal offsets broadened to include values from 0 to +5 nt, consistent with contributions from both binding modes. The offset preferences of miR-155 also broadened with increased pairing. However, instead of coinciding with pairing at position 11, these broadened preferences coincided with pairing to the G19-G20-G21-G22 stretch near the 3′ end of miR-155.

In summary, the most optimal 3′ sites each paired to at least two nucleotides of the miRNA segment spanning positions 13–16, which was previously identified as most consequential for 3′ pairing, but frequently did not pair to the entire segment. Shorter optimal sites consistently preferred pairing to G nucleotides adjacent to miRNA positions 13–16. For example, shorter optimal sites to let-7a paired to the G11-G12 sequence element 5′ of this segment rather than to G15-U16 (*Figure 2B*, right), the optimal 4-nt site to miR-1 paired to G12 rather than to U16 (*Figure 4A*, right), and intermediate-length optimal sites to miR-155 paired to G19-G20-G21 rather than to G13-U14 (*Figure 4B*, right). These trends were also observed when examining many combinations of positions, lengths, and offsets for miR-1 and miR-155 (*Figure 4—figure supplement 1*). In aggregate, these results supported the report of an intrinsic preference for pairing to miRNA nucleotides 13–16 (*Grimson et al., 2007*) but also indicated that the miRNA sequence imparts additional preferences, resulting in unanticipated differences between the optimal sites of individual miRNAs. These sequence-specific preferences tended to favor pairing to G residues of the miRNA, which was presumably explained by the greater stability of G:C pairing over A:U pairing, although the presence of only a single C nucleotide prevented investigation of whether pairing to G was preferred over pairing to C. We also observed differences between miRNAs in the strength of 3′ pairing. Compared to 3′-site affinities observed for let-7a, affinities were substantially lower for miR-1 and substantially higher for miR-155 (median increase in affinity with 11 bp of 3′ pairing of 36-fold, 5.8-fold, and 133-fold for let-7a, miR-1 and miR-155, respectively). Thus, our results indicated that association of the guide RNA with the AGO protein does not fully standardize either the architecture of optimal 3′ pairing or the magnitude of its benefit.

## Pairing and offset coefficients describe unique 3′-pairing profiles for each miRNA

To summarize the results for miR-1 and miR-155, we generated heat maps representing the binding affinity at all possible pairing positions for all pairing lengths of 4–11 bp, as a function of pairing offset (*Figure 4C and D*), as with let-7a (*Figure 2E*). The similarities observed between heat maps for the same miRNA at different offsets indicated that each change in offset altered the binding affinity of all 3′-pairing possibilities in a consistent manner, which in turn indicated that for each of the three miRNAs, the effect of pairing offset was largely independent of the effect of guide–target complementarity (*Figures 2E and 4C, D*). This overall independence was observed for let-7a, despite its two binding modes, because the contribution of the positive-offset binding mode, which had the higher affinities, dominated over that of the other binding mode.

To test this independence, we examined how well the affinities could be explained as the product of two coefficients, one representing the contribution of the pairing range, which was defined by pairing position and length (represented by the location of a cell within the heat maps of *Figures 2E*

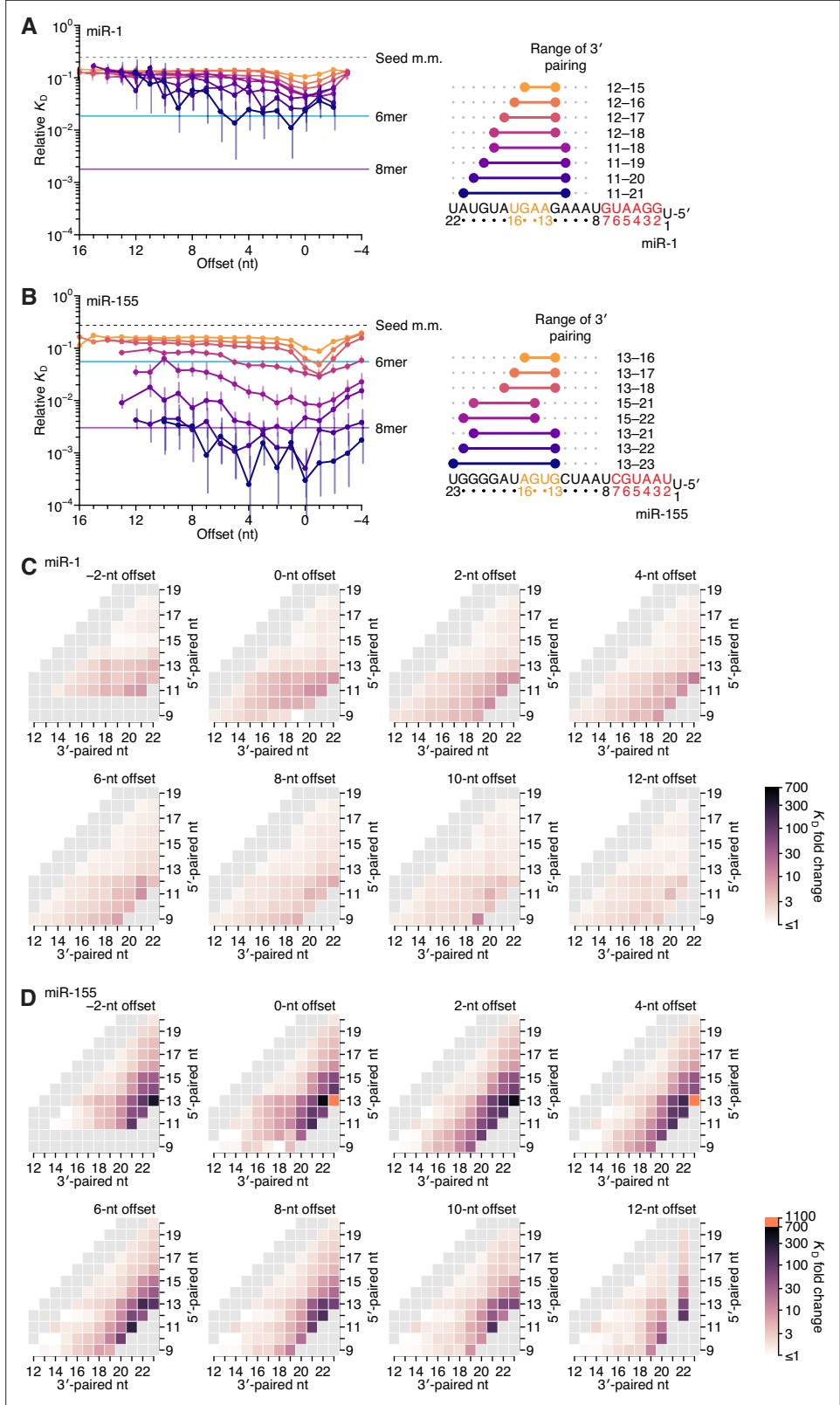

**Figure 4.** Relative affinity measurements of 3'-compensatory sites of miR-1 and miR-155. (**A**) Relative $K_D$ values of miR-1 3'-compensatory sites that had optimally positioned 3' pairing of 4–11 bp. Otherwise, this panel is as in **Figure 2B**. (**B**) Relative $K_D$ values of miR-155 3'-compensatory sites that had optimally positioned 3' pairing of

*Figure 4 continued on next page*

*Figure 4 continued*

4–11 bp. Otherwise, this panel is as in *Figure 2B*. (**C and D**) Affinity profiles of the 3′ regions of miR-1 (**C**) and miR-155 (**D**). Otherwise, these panels are as in *Figure 2E*.

The online version of this article includes the following figure supplement(s) for figure 4:

**Figure supplement 1.** Length, position, and offset trends of miR-1 and miR-155 indicate one binding mode.

*and 4C, D*), and the other representing the contribution of the pairing offset. Our model fit the data well ($r^2$ = 0.92, 0.86, and 0.96 for let-7a, miR-1, and miR-155, respectively; *Figure 5—figure supplement 1*), and yielded a set of pairing and offset coefficients for each miRNA. Each pairing coefficient represented the maximum beneficial ΔG associated with complementarity to the corresponding range of miRNA nucleotides, and each offset coefficient represented the fraction of the maximum beneficial ΔG observed at each pairing offset (*Figure 5A–C*). For each miRNA, the pairing coefficients corresponded well with the affinities observed at the preferred offset (*Figure 5A–C*, comparison of right-most heat maps; $r^2$ = 0.98, 0.97, and 0.96, respectively). Moreover, these coefficients, which distilled the pairing preferences indicated by the 934, 1061, and 1180 relative $K_D$ values measured for let-7a, miR-1, and miR-155, respectively, quantitatively captured the qualitative observations made earlier from analysis of subsets of the data.

Because the pairing coefficients represented the thermodynamic benefit of each pairing possibility, we examined how well each set of pairing coefficients was explained by the nearest-neighbor model that predicts the stability of RNA hybridization in solution. To do so, we calculated the predicted ΔG value for each 3′ site (*Figure 5D*) and adjusted each value by subtracting the mean value for that length of pairing, which was done to remove the trivial effect of increasing pairing length (*Figure 5E*). When comparing these length-adjusted values with analogously adjusted pairing coefficients, we observed a strong relationship for both let-7a and miR-155, and a much weaker relationship for miR-1. Nevertheless, even when focusing on results for let-7a and miR-155, the apparent effect size was less than that expected by the relationship ΔG = −RT ln$K$ (*Figure 5E*, dashed lines). Thus, as observed with the miRNA seed region (*Salomon et al., 2015*; *McGeary et al., 2019*), compared to RNA free in solution, association with AGO reduces the differences in binding energy observed when hybridizing to different miRNA 3′-end sequences.

This reduction in magnitude also applied to the overall contribution of 3′ pairing (*Figure 5—figure supplement 2*). For instance, although the >200-fold differences in binding affinity imparted by the top 11-nt 3′ sites of let-7a and miR-155 might seem large, the ΔG predicted for each of these sites was −14.8 kcal/mol and −20.1 kcal/mol, which corresponded to respective fold differences in affinity of $2.7 \times 10^{10}$ and $1.5 \times 10^{14}$. Presumably, the benefit of pairing to 3′ sites was mostly offset by the cost of disrupting favorable interactions between unpaired 3′ regions and AGO, as proposed in the context of fully paired sites (*Tomari and Zamore, 2005*). The magnitude of this inferred cost appeared specific to each miRNA, implying that AGO might have some sequence preferences when interacting with unpaired miRNA 3′ regions. For example, pairing to either nucleotides 9–19 of let-7a or nucleotides 11–21 of miR-1 was predicted to occur with equivalent ΔG values of −13.5 kcal/mol, yet the model-determined contributions of these sites were 160- and 14-fold, respectively (*Figure 5—figure supplement 2A*, left and middle).

Separating the comparison between $K_D$ fold-change and predicted ΔG based on whether the contiguous range of pairing included the G11, G12, and G20 of let-7a, miR-1, and miR-155, respectively, revealed a cooperative benefit of pairing to these nucleotides (*Figure 5—figure supplement 2B, C*), such that their inclusion within the 3′ pairing enabled the other paired nucleotides to contribute more to the interaction. We also note that using the measured affinities rather than pairing coefficients did not increase agreement with ΔG (*Figure 5—figure supplement 2D, E*), suggesting that the use of the pairing coefficients did not lead to loss of information contained within the data from which they were generated.

We next used data obtained previously from fully randomized libraries (*McGeary et al., 2019*) to extend our analyses to miR-124, lsy-6, and miR-7 (*Figure 5—figure supplement 3A–F*), for 3′ sites as long as eight nt. This upper-bound of 8 nt was selected because pairing and offset coefficients calculated for 3′ sites of let-7a, miR-1, and miR-155 using results from fully randomized libraries agreed with those calculated using results from the respective programmed libraries, provided that the sites did not exceed 8 nt (*Figure 5—figure supplement 3G, H*). Like let-7a, miR-124 had both preferred pairing

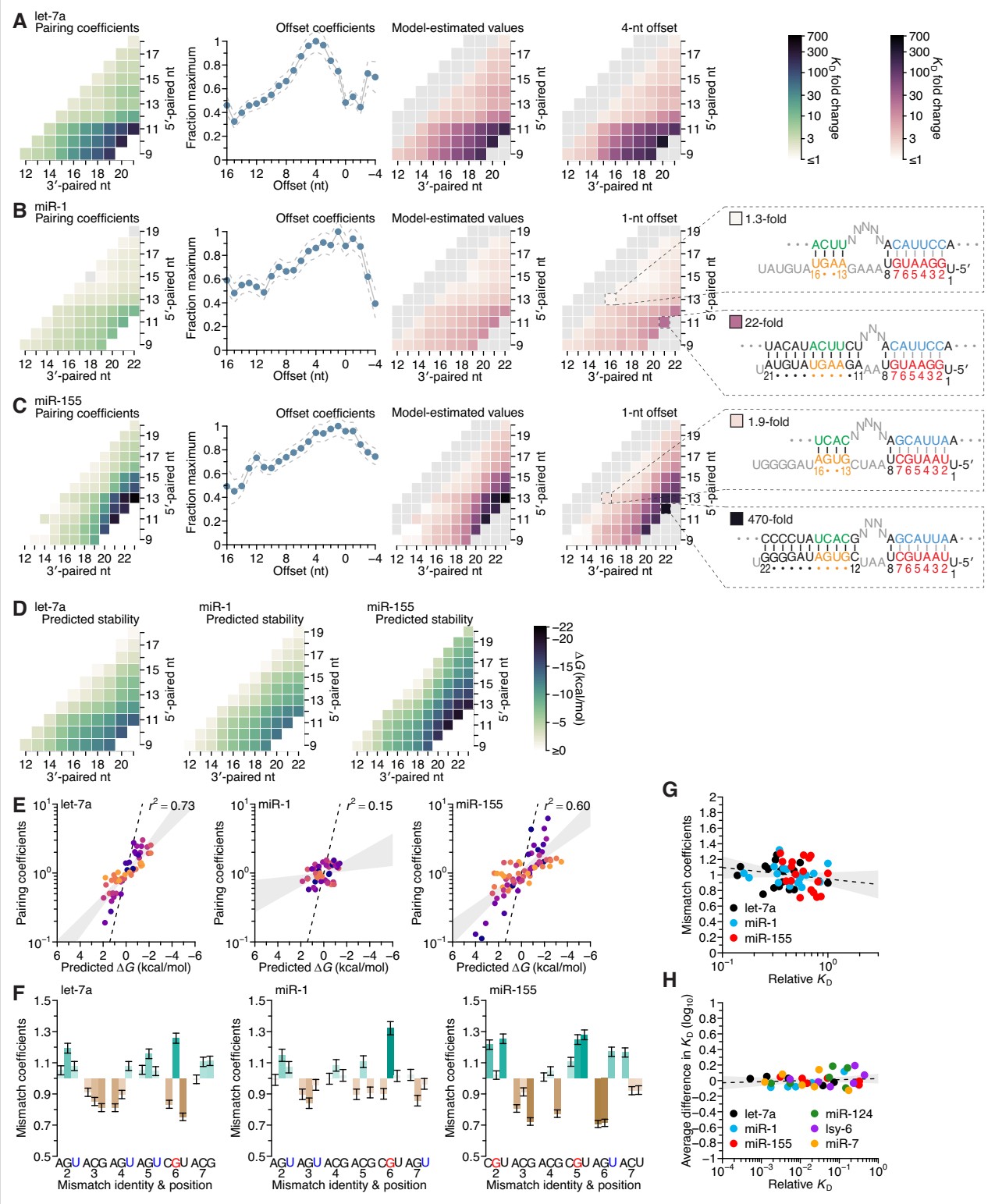

**Figure 5.** Distinct pairing-range, offset, and seed-mismatch preferences of different miRNAs. (**A–C**) Model-based analyses of 3'-pairing preferences of let-7a (**A**), miR-1 (**B**), and miR-155 (**C**). For each miRNA, 3'-pairing affinities are described by a set of pairing coefficients (left) and offset coefficients (middle-left; dashed lines, 95% confidence interval), which when multiplied together (middle-right) approximated measured $K_D$ fold-change values (right; let-7a values replotted from *Figure 2E*). The parameters were obtained by maximum-likelihood estimation with a nonlinear energy model. For both miR-1 (**B**) and miR-155 (**C**), the two pairing diagrams indicate the fold-change value and architecture for a 3' site pairing to miRNA nucleotides 13–16 (top) in comparison to the fold-change value and architecture of the 3' site with the greatest measured affinity (bottom) at their shared optimal offset of +1 nt.

*Figure 5 continued on next page*

*Figure 5 continued*

Pairing coefficients, model predictions, and $K_D$ fold-change values of miR-1 were not calculated for pairing to miRNA positions 15–18 and 19–22 because these two segments were identical (gray boxes). (**D**) Predicted $\Delta G$ values of the 3' sites with pairing ranges in (**A–C**). (**E**) The relationship between the model-derived pairing coefficients (**A–C**) and the predicted $\Delta G$ values (**D**). Points are colored according to pairing length, as in *Figure 2B*. To control for the trivial effect of increasing pairing length, pairing coefficients were divided by the geometric mean of all coefficients with the same length, and $\Delta G$ values of each length were normalized to the mean $\Delta G$ value of pairings with the same length. The gray region represents the 95% confidence interval of the relationship when fitting a linear model to the data ($r^2$, coefficient of determination), and the dashed line represents the predicted thermodynamic relationship given by $K = e^{-\Delta G/RT}$. (**F**) Distinct effects of seed mismatches on 3'-pairing affinities of let-7a, miR-1, and miR-155. For each miRNA, seed-mismatch coefficients were derived by maximum-likelihood estimation, fitting a nonlinear model to the $K_D$ fold-change values observed when examining 3'-site enrichment separately for each of the 18 seed mismatches. The error bars indicate 95% confidence intervals. Wobble pairing in which the G was in either the miRNA or the target is indicated in blue and red, respectively. (**G**) Relationship between affinity of 3'-compensatory pairing and that of seed-site binding. For each seed mismatch, the coefficient from (**F**) is plotted as a function of the relative $K_D$ value of that mismatch, as measured using results from the programmed libraries for let-7a (black), miR-1 (blue), and miR-155 (red). The dashed line shows the linear least-squares fit to the data, with the gray interval indicating the 95% confidence interval. (**H**) Relationship between affinity of 3'-supplementary pairing and that of seed-site binding. For each of the six seed-matched site types (*Figure 1A*, left) and for each of the six miRNAs (key), the relative affinity of the top quartile of all 4- and 5-nt 3' sites with their preferred offsets is plotted as a function of the relative affinity of the seed-matched site. Relative affinities were measured from analysis of previous AGO-RBNS that used a random-sequence library (*McGeary et al., 2019*; *Figure 5—figure supplement 8*).

The online version of this article includes the following figure supplement(s) for figure 5:

**Figure supplement 1.** Model prediction of 3'-compensatory pairing.

**Figure supplement 2.** Analysis of the correspondence of pairing coefficients with predicted $\Delta G$, and performance of seed mismatch–effect model.

**Figure supplement 3.** Distinct pairing and offset preferences of different miRNAs in random-sequence AGO-RBNS experiments.

**Figure supplement 4.** Identification of two binding modes for most miRNAs in random-sequence AGO-RBNS experiments.

**Figure supplement 5.** Model prediction of pairing, offset, and seed-mismatch effects for let-7a.

**Figure supplement 6.** Model prediction of pairing, offset, and seed-mismatch effects for miR-1.

**Figure supplement 7.** Model prediction of pairing, offset, and seed-mismatch effects for miR-155.

**Figure supplement 8.** Minimal influence of seed site type on 3'-supplementary pairing.

to position 11 and an optimal pairing offset of >2 nt (*Figure 5—figure supplement 3D*). To look for evidence of multiple binding modes, we repeated the analyses of both *Figure 2B* (for pairing lengths of 4–8 bp) and *Figure 2C* (for pairing lengths of 4 and 5 bp), using the prior AGO-RBNS data for miR-124, lsy-6, and miR-7 (*Figure 5—figure supplement 4*). For comparison, we also repeated these analyses using the prior AGO-RBNS data for let-7a, for which we had evidence of two binding modes from the programmed-library AGO-RBNS data. For each of the four miRNAs, we found evidence of the two binding modes. Both let-7a and miR-124 had the previously observed pattern, in which the positive-offset binding mode had binding affinity greater than that of the zero-offset binding mode (*Figure 5—figure supplement 4A–D*). However, lsy-6 and miR-7 had a different pattern, in which the binding affinities of both modes were similar (*Figure 5—figure supplement 4E–H*). Perhaps pairing to the G11-G12 dinucleotide found in both the let-7a and miR-124 sequences enabled the positive-offset binding mode to dominate over the zero-offset binding mode, whereas pairing to the single G11 found in lsy-6 and miR-7 added to site affinity but did not enable the positive-offset binding mode to dominate.

The analyses of miR-124 and lsy-6, which each had multiple C nucleotides in their 3' region, allowed us to return to the question of whether pairing to miRNA G nucleotides might be favored over pairing to C nucleotides. Pairing to C15 of lsy-6 substantially added to binding affinity. For example, the 4.2-fold greater affinity of the position 12–15 site over the position 11–14 site indicated that pairing to C15 was favored over pairing to G11, and extending pairing from positions 11–14 to 11–15 increased affinity 8.2-fold (*Figure 5—figure supplement 3E*). Pairing to C13 was also somewhat preferred, as illustrated by the 1.8-fold greater affinity of the 13–17 site over the 14–18 site, and the 3.2-fold benefit of extending pairing from positions 14–18 to 13–18. However, pairing to C19-C20 of miR-124 did not seem to have the same impact as pairing to G19-G20 of miR-155, as illustrated by the negligible (0.9-fold) benefit of extending the miR-124 pairing from positions 13–18 to 13–20, compared to the 14-fold benefit for miR-155. These results supported the idea that pairing to a G in the miRNA 3' region is generally favored over pairing to a C, although pairing to a C located within positions 13–16 of the 3' region can be impactful.

## The type of seed mismatch affects the affinity of 3′ pairing

To examine the influence of seed-mismatch position and identity, we analyzed the full set of 16,235, 18,076, and 19,666 relative $K_D$ values of let-7a, miR-1, and miR-155, no longer combining read counts for the 18 possible seed-mismatch sites in the programmed library prior to $K_D$ estimation. For each pairing, offset, and seed-mismatch possibility, the relative $K_D$ value of the 3′-compensatory site was divided by that of its seed-mismatch site to generate a fold-change value representing the contribution of the 3′ site to affinity (*Figure 5—figure supplements 5–7*). These values revealed a striking effect of seed-mismatch identity on the benefit of 3′ pairing. This effect was of greater magnitude for more favorable 3′ sites, causing affinities to vary >10-fold for the most optimal sites to miR-155. To further study this effect, we expanded our model to include a seed-mismatch coefficient, such that each $\log_{10}(K_D$ fold change) value was described as the product of the pairing, offset, and seed-mismatch coefficients corresponding to its 3′-pairing architecture (*Figure 5—figure supplements 5–7*).

The affinity of seed-mismatch sites lacking 3′ pairing had little relationship with the influence of the mismatch on 3′-pairing affinity (*Figure 5G*). Likewise, examination of data from the six random-library AGO-RBNS experiments found no relationship between the affinities of canonical sites lacking 3′ pairing and the relative influence of each canonical site on the benefit of supplemental pairing (*Figure 5H* and *Figure 5—figure supplement 8*). Furthermore, the average effect of canonical-site type on 3′ binding affinity was

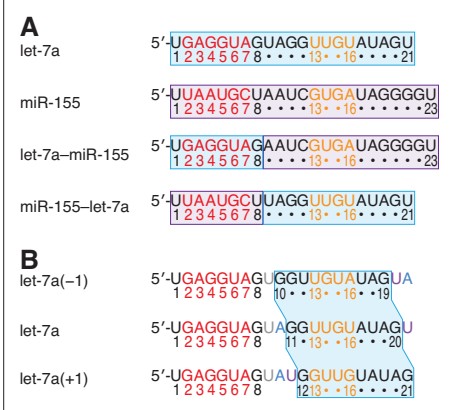

**Figure 6.** Variant miRNAs designed to query the contributions of the seed and 3′ regions to binding, and the positional dependence of pairing preferences of the 3′ region. (**A**) Sequences of native let-7a, native miR-155, a chimeric miRNA containing the seed region of let-7a appended to nucleotides 9–23 of miR-155 (let-7a–miR-155), and a chimeric miRNA containing the seed region of miR-155 appended to nucleotides 9–21 of let-7a (miR-155–let-7a). (**B**) Sequences of let-7a(−1), which has a 3′ region permuted one nucleotide toward the 5′ end, native let-7a, and let-7a(+1), which has a 3′ region permuted one nucleotide toward the 3′ end. The 3′ sequence shared between all three miRNAs is shaded in blue, and the A and U nucleotides that were rearranged to generate the permuted variants are in blue and purple, respectively.

The online version of this article includes the following figure supplement(s) for figure 6:

**Figure supplement 1.** Independence of seed-mismatch and 3′-sequence effects.

**Figure supplement 2.** Sequence preferences for 3′ sites are maintained at adjacent positions.

small, with only six out of the 36 miRNA–site combinations having a >0.1 effect on $\log_{10}(K_D$ fold change), corresponding to an ~25% change in binding affinity (*Figure 5H*). Together, these results indicate that for 3′-supplementary pairing, the benefit of the 3′ pairing is largely the same between sites, but that for 3′-compensatory pairing, the potential benefit of 3′ pairing differs depending on the identity of the seed mismatch. This might be due to a differential ability of these mismatches to elicit a conformational change in AGO allowing pairing to the 3′ end (*Schirle et al., 2014*; *Sheu-Gruttadauria et al., 2019b*). Another potential contribution might stem from variation in elemental rate constants of seed-mismatch sites of similar affinity, whereby some sites have dwell times that are too short to establish pairing to the miRNA 3′ region.

When comparing the effects for guide–target nucleotide possibilities, strong trends did not emerge within miRNAs (e.g., when comparing the effects of mismatches to the G at position 2 with those of the mismatches to the G at position 4 of let-7a), or between miRNAs (e.g. when comparing the effects of mismatches to the G at position 3 of miR-1 with those to the G at position 6 of miR-155) (*Figure 5F*). However, in cases in which the same nucleotide occurred at the same position for two different miRNAs, some correspondence was observed (positions 2 and 6 of let-7a and miR-1, position 3 of let-7a and miR-155, position 4 of miR-1 and miR-155) (*Figure 5F*). Notably, the miRNA–target U:G mismatch at position 6, which was the most favored mismatch for both let-7 and miR-1, occurs within one of the two compensatory sites within the 3′ UTR of *C. elegans lin-41* (*Figure 3A*; *Pasquinelli*

*et al., 2000*; *Reinhart et al., 2000*), further helping to explain the activity of this site in *C. elegans* development.

## Pairing preferences of miRNA 3'-end nucleotides are independent of the seed sequence and maintained at adjacent positions

Having found that the 3'-pairing affinities of each of the three miRNAs were largely a function of the miRNA pairing, offset, and seed-mismatch preferences, we investigated the extent to which different regions of each miRNA contributed to these preferences. To do so, we performed AGO-RBNS with synthetic miRNA variants. Two of these variants were chimeric miRNAs in which nucleotides 1–8 of let-7a and miR-155 were swapped (*Figure 6A*). The other two were let-7a variants in which the 3' sequence was shifted by one nucleotide in either direction (*Figure 6B*). Results for these variants showed that both the pairing and offset preferences tracked with the 3' sequence (*Figure 6—figure supplement 1A–G*), with the quantitative contribution of each 3' nucleotide to pairing largely maintained when shifted to an adjacent position (*Figure 6—figure supplement 2A–E*). By contrast, the seed-mismatch preferences tracked with the seed sequence, with little pairwise difference in these preferences observed for miRNA variants sharing nucleotides 1–8 but possessing distinct 3' sequences (*Figure 6—figure supplement 1H, I* and *Figure 6—figure supplement 2F*).

## Effects of mismatches within 3' sites are consistent across miRNAs but explained poorly by the nearest-neighbor model

Having systematically analyzed the effects of seed-mismatch identity and of the length, position, and offset of perfect 3' pairing, we next sought to measure the effects of any imperfections—that is, mismatches, wobbles, or bulged nucleotides—within this 3' pairing. Accordingly, we measured the relative affinities of variants of each site considered thus far, looking at each possible variant that had one of the eight possible single-nucleotide imperfections at one position within the site. These eight imperfections considered at each position of interest included three possible mismatched nucleotides (including G:U wobbles), four possible single-nucleotide bulges (occurring opposite the linkage of two miRNA positions and assigned to the more 3' miRNA position), and one single-nucleotide deletion (i.e., a bulged nucleotide in the miRNA). Consideration of these variants together with the original sites with perfect contiguous pairing resulted in the measurement of $K_D$ values for 38,108 let-7a sites, 44,190 miR-1 sites, and 52,166 miR-155 sites. Analysis of these variants in the context of the best sites at each length (*Figure 7A–C* and *Figure 7—figure supplement 1*) revealed no imperfections that increased 3'-site affinity, which indicated that there were no positions at which the altered helical geometry of a mismatch was favored over Watson–Crick pairing. When comparing effects of internal mismatches to those of mismatches occurring at the end of the pairing, no striking differences were observed. Nonetheless, effects at some positions were more striking than others, with larger effects observed for mismatches involving any of nucleotides 11–15 of let-7a (*Figure 7A*), 12–15 of miR-1 (*Figure 7B*), and 15–22 of miR-155 (*Figure 7C*), which concurred with the importance of extending pairing to G11-G12, G12, and G19-G20-G21-G22 of the respective miRNAs.

To investigate mismatch tolerance across the range of miRNA 3'-end positions, we calculated the geometric mean of the $K_D$ fold change for a mismatch at each position for all three miRNAs, averaging both over the three mismatches at each position and over each of the 10-nt sites that contained the position (*Figure 7D*). As expected, reduced binding affinity tracked with the importance of the positions for 3' pairing, with the greatest effects observed at G11 and G12 of let-7a, the G12–G15 of miR-1, and G13 and G15–G21 of miR-155 (*Figure 7D*). The greater importance of pairing to G13 compared to pairing to C12 of miR-155 further supported the idea that pairing to G had a greater impact over pairing to C in the miRNA 3' region. Nonetheless, extending the analyses of mismatches, wobbles, and bulges to the random-sequence RBNS datasets previously acquired for six miRNAs (*Figure 7—figure supplement 2*) indicated that disrupting pairing to either C13 or C15 of the C13-G14-C15 trinucleotide of lsy-6 greatly reduced affinity. Thus, in some nearest-neighbor and positional contexts, pairing to a miRNA C nucleotide can be as important as pairing to a miRNA G nucleotide. More generally, these results showed that the effect of a mismatch to a particular nucleotide was informed primarily by the overall importance of that miRNA nucleotide for pairing (as determined by its nucleotide identity and position within the miRNA 3' end), irrespective of whether the target nucleotide fell within the middle or terminus of the 3' site.

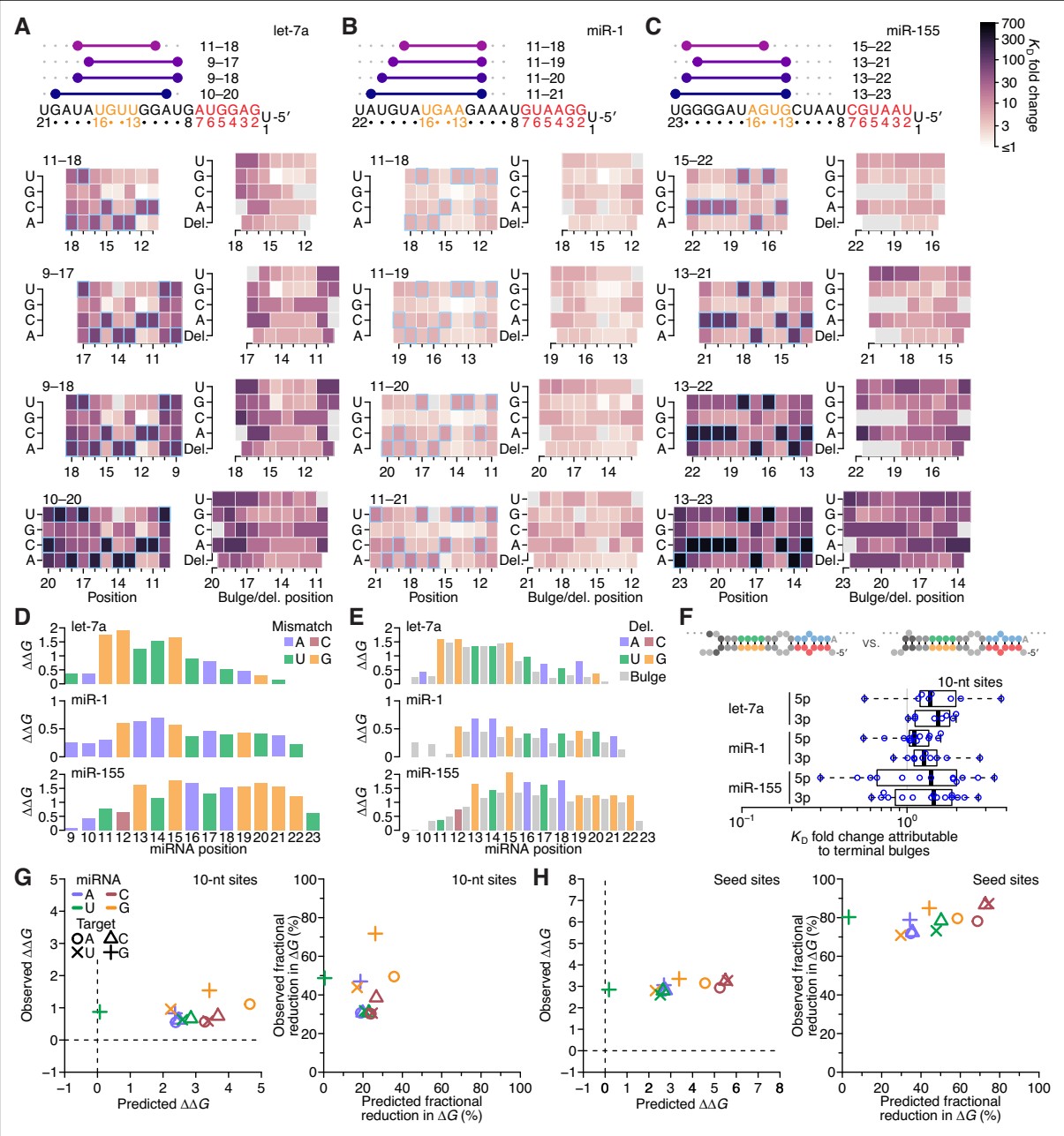

**Figure 7.** The impact of mismatched, bulged, and deleted target nucleotides on 3′-compensatory pairing. (**A**) The effect of mismatched, bulged, and deleted target nucleotides on 3′-compensatory pairing to let-7a. At the top is a schematic depicting the position of highest-affinity 3′-pairing for 3′ sites of lengths 8–11 nt, redrawn from **Figure 2B**. Below, at the left are heat maps corresponding to each of the pairing positions shown above, indicating the affinities with each of the four possible nucleotides at each position of the site. Cells corresponding to the Watson–Crick match are outlined in blue. Cells for affinities of mismatches that could not be calculated due to sequence similarity to another site type (e.g., the mismatched U across from position 14, which was indistinguishable from a 6mer-m8 seed site) are in gray. To the right are heat maps that correspond to the same pairing ranges but indicate the effects of a bulged or a deleted (del.) 3′-target nucleotide. A bulged nucleotide at position *n* corresponded to an extra target nucleotide inserted between the nucleotides pairing to miRNA positions *n* − 1 and *n*. (**B and C**) The effects of mismatched, bulged, and deleted target nucleotides on 3′-compensatory pairing to miR-1 (**B**) and miR-155 (**C**). Otherwise, these panels are as in (**A**). (**D**) Profiles of 3′-pairing mismatch tolerances. Each bar represents the ΔΔ*G* value when averaging over the three possible mismatches at that position, for let-7a (top), miR-1 (middle), and miR-155 (bottom). Each of the mismatch ΔΔ*G* values was an average of the values observed in the context of each 10-nt 3′ site that included the position. The color indicates whether the miRNA nucleotide was an A (blue), U (green), C (red), or G (yellow). (**E**) Profiles of tolerances to bulged and deleted nucleotides. Each colored bar represents the ΔΔ*G* value when deleting the target nucleotide complementary to the miRNA at that position, and each dark gray bar represents the ΔΔ*G* value when averaging all four bulged nucleotide possibilities occurring at that inter-nucleotide position, for let-7a (top), miR-1 (middle), and miR-155 (bottom). Each of the ΔΔ*G* values represents the average of the values observed in the context of each 10-nt 3′ site

*Figure 7 continued on next page*

*Figure 7 continued*

that included the position. The color of each bar corresponding to a deletion indicates whether the resulting bulged miRNA nucleotide was an A (blue), U (green), C (red), or G (yellow). (**F**) The tolerance of bulged nucleotides near the ends of 3′ sites. Plotted are ratios of $K_D$ fold-changes comparing a site that has a bulged nucleotide between the penultimate and terminal base pairs with a site that does not have the terminal base pair (in which case, the bulged nucleotide in the former pairing architecture becomes a terminal mismatch). The box plots indicate the minimum, lower-quartile, median, upper-quartile, and maximum values. For each of the three miRNAs, comparisons are made for bulges occurring at the 5′ end of the 3′ pairing (5p), and at the 3′ end of the 3′ pairing (3p). The vertical gray line indicates a $K_D$ fold-change ratio of 1.0. At the top is an example of a 3p comparison. (**G**) Comparison of the measured mismatch ΔΔG values in 3′ sites with values predicted by nearest-neighbor rules. Left, comparison of the average measured ΔΔG value with the average predicted value for each of the 12 possible miRNA–target mismatch combinations. Right, comparison of measured and predicted average fractional reduction in ΔG attributed to each mismatch. The fractional reduction was given by ($\Delta G_{WC} - \Delta G_{mm}$)/$\Delta G_{WC}$, where $\Delta G_{WC}$ corresponds to the ΔG of the site with full Watson–Crick pairing, and $\Delta G_{mm}$ corresponds to the ΔG of a site containing the mismatch. These average values were calculated using $K_D$ fold-change values determined for 10-nt sites, first averaging results for the same position over all 10-nt sites that included the position, then averaging results for that mismatch across all positions of the miRNA that had that mismatch, and then averaging the results across all three miRNAs. Colors and symbols indicate miRNA and target nucleotide identities, respectively (key). (**H**) Comparison of the measured seed mismatch ΔΔG values with values predicted by nearest-neighbor rules. For each mismatch type, both the measured and predicted ΔΔG values were the average over all occurrences within positions 2–7 for let-7a, miR-1, miR-155, miR-124, lsy-6, and miR-7, using relative $K_D$ values from analyses of random-sequence AGO-RBNS results. Otherwise, this panel is as in (**G**).

The online version of this article includes the following figure supplement(s) for figure 7:

**Figure supplement 1.** Further analyses of the impact of mismatched, bulged, and deleted target nucleotides on 3′-compensatory pairing.

**Figure supplement 2.** The impact of mismatched, bulged, and deleted target nucleotides on 3′-compensatory pairing in random-sequence AGO-RBNS experiments.

To summarize the positional tolerance of bulges, we averaged the effects of the four bulges at each inter-nucleotide position and over each of the 10-nt sites containing that position. Likewise for the deletions, we averaged each single possibility over the 10-nt sites containing that position (*Figure 7E*). At each position, the severity of both types of lesions tracked with that observed for the mismatches, with the effects of deletions generally similar to those of their corresponding mismatches, and effects of bulged nucleotides marginally less severe.

To examine if the benefit of bulged nucleotides over mismatched nucleotides applied to the very 5′ and 3′ ends of 3′ sites, we considered all possible 10-nt sites for all three miRNAs with programmed libraries, and calculated the fold difference in relative $K_D$ observed when comparing a site with a terminal mismatch to that of the site with a corresponding terminal bulged nucleotide (i.e. the site variant in which the target nucleotide following the mismatch can pair to the mismatched miRNA nucleotide). For each miRNA, a small but significant benefit to terminal bulges was observed (*Figure 7F*; p = $2.4 \times 10^{-5}$, $1.4 \times 10^{-6}$, and $4.5 \times 10^{-4}$ for let-7a, miR-1, and miR-155, respectively; one-tailed Wilcoxon signed rank test). Thus, an isolated complementary target nucleotide separated from a longer contiguous stretch of pairing can contribute modestly to site affinity.

To enable comparison of the observed effects of mismatches with those predicted by the nearest-neighbor model of RNA duplex stability, we calculated the ΔΔG of each mismatch in the context of all 10-nt 3′ sites of the three miRNAs. We first averaged these values over all the contiguous sites, and then over all positions with the same miRNA nucleotide, and then over the three miRNAs, resulting in one global average ΔΔG value for each of the 12 possible miRNA–target mismatch possibilities. Comparison of these values with those predicted using the nearest-neighbor parameters revealed that the effects of the mismatches were typically much lower than expected for RNA in solution, with no strong relationship between the observed and predicted ΔΔG values (*Figure 7G*, left; $r^2 = 0.02$). The outlier in this analysis was the miRNA–target U:G wobble, which was as disruptive as the typical mismatch but predicted to be much less so (*Figure 7G*, left, green +). Next, to account directly for the reduced binding energy of the fully complementary sites in comparison to their predicted ΔG values, we compared the average observed and predicted fractional reduction in ΔG of each site caused by each of the 12 mismatch values (*Figure 7G*, right). For eight of 12 mismatches, the fractional reduction in ΔG was within 10% of its prediction, but the miRNA–target A:G, G:G, G:U, and U:G mismatches respectively caused 31%, 42%, 21%, and 48% more reduction in binding energy than predicted. These results indicated that the nearest-neighbor parameters were not suited for predicting the contribution of miRNA 3′ pairing in three respects: (1) the overall contribution to binding energy was far less than that predicted, (2) mismatched target G nucleotides were relatively more deleterious than predicted, and (3) wobble pairing was relatively less favorable than predicted. Indeed, the U:G possibility, which

both contained a target G nucleotide and was a wobble, was the mismatch with the greatest deviation from expectation.

For comparison, we repeated these analyses for mismatches to the miRNA seed (i.e., miRNA positions 2–7) within the context of canonical 8mer pairing, calculating the average $\Delta\Delta G$ and the fractional reduction in $\Delta G$ for each type of mismatch for each of the six miRNAs for which there was random-sequence RBNS data (*McGeary et al., 2019*; *Figure 7H*). These analyses indicated that the effects of mismatches within seed pairing also did not agree with predicted pairing energetics, albeit differently than the effects of mismatches within 3′ pairing. First, a mismatch within the seed pairing had a much larger influence on $\Delta\Delta G$ than did a mismatch within the 3′ pairing. Moreover, the reductions in binding affinities for mismatches within the seed pairing were even more regular than those for mismatches within the 3′ pairing, with a ~3 kcal/mol detriment for each of the 12 mismatch/wobble possibilities (*Figure 7H*, left). The fractional reduction in $\Delta G$ had a similarly large and uniform effect size, with no subset of the mismatch possibilities showing a relationship with that predicted (*Figure 7H*, right). Thus, the binding preferences at both the seed and 3′ regions of the miRNA were not well explained by nearest-neighbor rules, although the nature of the deviations differed in these two regions.

## Discussion

An AGO-loaded miRNA can be divided into three regions: the seed region (nucleotides 2–8), the central region (nucleotides 9–10 or 9–11), and the 3′ region (*Figure 1A*; *Bartel, 2018*). Because the most effective 3′ pairing is reported to center on nucleotides 13–16 (*Grimson et al., 2007*), some subdivide the 3′ region into the 3′-supplementary region (nucleotides 13–16), and the tail (nucleotides 17 to the terminus), while expanding the central region to include nucleotide 12 (*Wee et al., 2012*; *Schirle et al., 2014*; *Salomon et al., 2015*; *Sheu-Gruttadauria et al., 2019b*). The structure of AGO2–miR-122 bound to a 3′-supplementary site, which shows that miRNA nucleotides 9–11 are not available for pairing due to both helical distortion and inaccessibility caused by residues of the PIWI and L2 loop, seems to support the notion of a 3′-supplementary region at nucleotides 13–16 (*Sheu-Gruttadauria et al., 2019b*). However, greater affinities are observed with more extended 3′ pairing (*Becker et al., 2019*; *Sheu-Gruttadauria et al., 2019a*), and we found that 3′-site affinities nearly always increased as potential for pairing expanded to include most of the 3′ region—and in the positive-offset binding mode, some of the central region. Thus, productive 3′ pairing can encompass the entire miRNA 3′ region and should not be thought of as limited to a short 3′-supplementary region. Indeed, the study reporting that pairing to nucleotides 13–16 is most effective for supplementing seed pairing uses a model for predicting the efficacy of 3′ pairing that rewards extension of that pairing into the remainder of the 3′ region (*Grimson et al., 2007*).

Also problematic for the notion of a short 3′-supplementary region common to all miRNAs was our observation that the positions most important for 3′ pairing differed between different miRNAs. For example, at their optimal offsets, both let-7a and miR-124 preferred pairing to nucleotides 11–14 over pairing to nucleotides 13–16 (*Figures 2B and 5A*, and *Figure 5—figure supplement 3A,D*), and the synthetic let-7a(−1) preferred pairing to nucleotides 10–13 over pairing to nucleotides 13–16 (*Figure 6—figure supplement 2B*). Moreover, although miR-155 preferred pairing to nucleotides 13–16 over other 4-nt possibilities, when examining 7-nt 3′ sites, it preferred pairing to nucleotides 15–21 over pairing that included nucleotides 13–16 (*Figure 4B* and *Figure 4—figure supplement 1B*). These observations showing that the preferred positions of 3′ pairing can vary so widely between miRNAs, to include virtually any nucleotide downstream of the seed, argued strongly against assigning the same short 3′-supplementary region to all miRNAs.

Our observations that pairing to nucleotides 11–14 of let-7a imparted greater affinity than did pairing to nucleotides 13–16 (*Figure 2B and C*) and that pairing to nucleotides 11–19 imparted greater repression than did pairing to nucleotides 13–21 (*Figure 3*), concurred with recent analyses of the relative importance of these nucleotides in *Caenorhabditis elegans. C. elegans* requires *let-7* repression of *lin-41* for viability, and this repression occurs through two 3′-compensatory sites that each have pairing to nucleotides 11–19 of the miRNA (*Figure 3A*; *Pasquinelli et al., 2000*; *Reinhart et al., 2000*; *Aeschimann et al., 2019*). Mutagenesis of individual nucleotides of the *let-7* miRNA indicates that nucleotides 11, 12, and 13 are each critical for viability, whereas nucleotides 14, 15, and 16 each have intermediate importance, and nucleotides 17, 18, and 19 each have no detectable importance (*Duan et al., 2021*). Inspection of our data for let-7a, examining the effects of mismatches

within the 3′ site that has the same architecture as that of the two sites within *lin-41* (i.e., 9 bp of pairing beginning at position 11 with an offset of +1 nt) revealed a similar polarity, with mismatches near position 11 tending to be most consequential and those near position 19 tending to be least consequential (*Figure 7—figure supplement 1D*).

Although our results showed that preferred pairing often did not correspond precisely to positions 13–16, preferred pairing did always at least partially overlap this segment. Moreover, as pairing lengths increased from 4 to 6 bp, overlap between preferred pairing and this segment increased, such that the preferred 6-nt sites for let-7a, miR-1, miR-155, miR-124, miR-7 and lsy-6 each included pairing to miRNA nucleotides 13–16. The only exception we observed was the preferred 6-nt site for synthetic let-7a(−1), which paired to nucleotides 10–15. Thus, our results explain why an overall preference for pairing to nucleotides 13–16 was detected in meta-analyses of both functional data for 11 miRNAs and evolutionary conservation of sites for 73 miRNA families (*Grimson et al., 2007*). Our key added insight is that sequence identity in the 3′ region—particularly the placement of stretches of G residues—imparts additional preferences that supplement the positional preferences to specify different optimal regions of 3′ pairing for different miRNAs.

Another key insight is evidence of two distinct 3′-binding modes, observed as different offset preferences of let-7a, miR-124, lsy-6, and miR-7 with and without pairing to nucleotide 11 (*Figure 2B and C* and *Figure 5—figure supplement 4*). In the zero-offset binding mode, an offset of 0 or +1 nt is optimal for 3′ pairing starting at position 12, whereas in the positive-offset binding mode, additional nucleotides are required to bridge pairing to positions 10 or 11, resulting in optimal offsets that exceed +1 nt. In a crystal structure of AGO2–miR-122 bound to a 3′-supplementary target that pairs to nucleotides 13–16 with an offset of 0 nt, nucleotide 12 is the first nucleotide available for pairing, whereas pairing to nucleotide 11 is occluded by the central gate (*Sheu-Gruttadauria et al., 2019b*). We suggest that this structure reflects the conformation of the zero-offset binding mode, as it provides a physical model for why extension of potential pairing from nucleotide 12 to nucleotide 11 results in almost no increased binding affinity (*Figure 5—figure supplement 4*) for sites with an offset of 0 nt. However, another structure will be required to visualize the positive-offset binding mode that enables optimal pairing to let-7a and miR-124, as well as strong pairing to lsy-6 and miR-7. Genetically identified sites inferred to be utilizing this second binding mode include the two let-7a sites within the 3′ UTR of *C. elegans lin-41*, which both include pairing to nucleotide 11 and an offset of +1 nt, as well as the first lsy-6 site within the 3′ UTR of *C. elegans cog-1*, which includes pairing to nucleotide 11 and an offset of +2 nt. The discovery of these two binding modes required knowledge of the interplay between preferred pairing position and preferred pairing offset, which underscored the utility of obtaining affinity measurements for a large diversity of 3′ sites.

Early attempts to either explain targeting efficacy or predict target sites used scores incorporating, among other things, the predicted binding energy between the miRNAs and their proposed targets (*Enright et al., 2003*; *Lewis et al., 2003*; *Doench and Sharp, 2004*; *Rajewsky and Socci, 2004*; *Krek et al., 2005*). That these metrics were less useful in identifying consequential 3′ pairing than simpler rubrics scoring only the length and position of complementarity (*Grimson et al., 2007*) suggests that the parameters derived from interactions of purified RNAs in solution are not directly relevant to miRNAs associated with AGO. The breadth of our affinity measurements provided the ability to assess why such parameters are not as useful. Although high correspondence was observed between the predicted $\Delta G$ and measured 3′-pairing affinities (*Figure 5—figure supplement 2A*), for miR-1 this relationship nearly disappeared when normalizing for pairing length (*Figure 5E*). For let-7a and miR-155, a relationship was retained after normalizing for length, but four factors limit the utility of using this relationship for ranking target predictions. The first is the strong effect of position, with complementarity to the seed much more consequential than complementarity to the 3′ region, and complementarity at some positions in the 3′ region more consequential than complementarity to others, and much more consequential than complementarity to positions 1, 9, and often, 10. The second is the effect of primary sequence, as illustrated by the outsized benefit pairing to the G11, G12, and G20 nucleotides of let-7a, miR-1, and miR-155, respectively (*Figure 5—figure supplement 2B, C*). The third is the poor relationship between the predicted and measured effects of some internal mismatches and wobbles (*Figure 7G*), and the fourth is a lack of a consistent relationship between predicted $\Delta G$ and measured binding affinities between miRNAs (*Figure 5—figure supplement 2A*, comparing the slope for miR-1 with that for either let-7a or miR-155).

Comparison of the 3′ regions of the four miRNAs that were more effective at 3′ pairing with those of the two that were not suggested a feature that might have conferred higher 3′-pairing affinity: the presence of two or more adjacent G nucleotides (e.g. the G11-G12 of both let-7a and miR-124, and the G19-G20-G21-G22 of miR-155). Although lsy-6 did not have an oligo(G) stretch, it did have a well-positioned C13-G14-C15 trinucleotide, which together with G11 was critical for pairing affinity. When considering all four miRNAs together, as well as the lack of any GG, CG, or GC dinucleotides within the 3′ regions of miR-1 or miR-7, we suggest that miRNAs with GG, CG, or GC dinucleotides within positions 13–16 are the ones most likely to participate in productive 3′ pairing, and that pairing that extends to an oligo(G) sequence outside of positions 13–16 preferentially enhances affinity.

The importance of pairing to miRNA G nucleotides, and not C nucleotides (other than the C13-G14-C15 of lsy-6), suggested that a miRNA–target G:C base pair is read out differently than a C:G base pair. Perhaps G nucleotides participate in base-stacking interactions that position or pre-organize the guide strand to favor nucleation of 3′ pairing. Alternatively, the explanation might involve target-site accessibility. Pairing to a C in the miRNA 3′ region would require a G in the vicinity of the seed match, which compared to a C would cause poorer target-site accessibility (*McGeary et al., 2019*), thereby reducing the net contribution to binding.

Our results also revealed a functional difference between 3′-supplementary and 3′-compensatory pairing. The added affinity of a 3′ site was relatively constant when it supplemented different sites that had seed matches (*Figure 5H* and *Figure 5—figure supplement 8*), whereas it varied in the context of different 3′-compensatory sites that had different seed mismatches (*Figure 5F* and *Figure 5—figure supplements 5–7*). The effects of seed mismatches were miRNA-specific and unrelated to their binding affinities (*Figure 5G*). Additionally, our experiments using chimeric miRNAs demonstrated the separability of the mismatch effects from the length, position, offset, and nucleotide-identity preferences of the 3′ region (*Figure 6—figure supplement 1*).

Pairing to the miRNA 3′ region not only increases site affinity and target repression, but it can also influence the stability of the miRNA itself, in a process called target-directed miRNA degradation (TDMD) (*Ameres et al., 2010*; *Cazalla et al., 2010*; *de la Mata et al., 2015*; *Bitetti et al., 2018*; *Kleaveland et al., 2018*). The handful of target sites known to trigger TDMD have diverse 3′-pairing architectures. For example, degradation of miR-7 triggered by the cellular Cyrano transcript occurs through a canonical 8mer site supplemented with 14 contiguous pairs to the 3′ end of the miRNA (*Kleaveland et al., 2018*), whereas degradation of miR-27a triggered by the m169 RNA from murine cytomegalovirus occurs through a canonical 7mer-A1 site supplemented with only six contiguous pairs to the 3′ end of the miRNA (*Marcinowski et al., 2012*). Our finding that miR-7 has the weakest 3′ pairing among the six miRNAs we studied provides a potential explanation as to why its TDMD trigger Cyrano has such a long 3′ site.

The crystal structures of several known TDMD substrates bound to their corresponding TDMD-inducing target sites reveal a distinct conformation for these AGO–miRNA–target RNA ternary complexes in comparison to ternary complexes that have supplementary pairing involving only nucleotides 13–16 (*Sheu-Gruttadauria et al., 2019a*; *Sheu-Gruttadauria et al., 2019b*). During TDMD, this distinct conformation is thought to be recognized by the ZSWIM8 E3 ubiquitin ligase, causing AGO proteolysis through the ubiquitin–proteasome system, which exposes the miRNA to degradation by cellular nucleases (*Han et al., 2020*; *Shi et al., 2020*). Our discovery of the two 3′ binding modes raises the question of whether one of them might be more compatible with TDMD, perhaps due to a preference of the ZSWIM8 E3 ligase. Although the TDMD ternary complexes of the published structures all have 3′ pairing beginning at nucleotide 12 or later and offsets of 0 or −1 nt (*Sheu-Gruttadauria et al., 2019a*) and thereby represent the zero-offset binding mode, the 3′ pairing between miR-7 and Cyrano begins at G11 and has a +2-nt offset, which represents the positive-offset binding mode. Thus, the two 3′-binding modes both appear to be compatible with either of the two gene-regulatory processes that involve 3′ pairing—TDMD and miRNA-mediated repression.

## Materials and methods

### Cell lines

All cells were cultured in DMEM (VWR, 45000–304) with 10% FBS (TaKaRa, 631106) at 37°C and with 5% $CO_2$. HEK293T cells (*McGeary et al., 2019*) were obtained from Bartel lab stocks and had tested negative for Mycoplasma. F9 cells were purchased directly from the ATCC.

### Purification of AGO2–miRNA complexes

AGO2–miRNA complexes were generated and purified as described (*McGeary et al., 2019*).

### Preparation of programmed RNA libraries

For each miRNA, a programmed library was constructed by in vitro transcription of multiple chemically synthesized DNA libraries, and then mixing these RNA libraries after gel purification. Each library contained 25 nt of entirely randomized sequence, followed by an 8-nt programmed site, followed by either 5 nt of random sequence (let-7a and miR-1 libraries) or 4 nt of random sequence (miR-155 library). For every AGO-RBNS experiment other than that with native miR-155, the final library was made by mixing six different libraries, in which each of the six libraries had at its programmed site an 8mer containing a mismatch to one of the six seed nucleotides. For the experiment with native miR-155, the library was assembled by mixing the six 8mer-mismatch libraries with a seventh library that had a 6mer at the programed site, flanked by non-A residues at positions 1 and 8.

Each individual library was transcribed from synthetic DNA (IDT) and purified as described (*McGeary et al., 2019*). The fraction of reads representing each of the 18 seed-mismatch sites, expected to be 5.6%, was measured by sequencing of each input library and found to be 3.4–8.0% for let-7a rep. 1, 2.9–8.7% for let-7a rep. 2, 3.3–7.8% for miR-1, 2.2–6.3% for miR-155, 3.4–8.0% for let-7a(+1) and let-7a(−1), 3.3–7.6% for let-7a–miR-155, and 2.6–6.1% for miR-155–let-7a.

### AGO-RBNS

AGO-RBNS was performed as described (*McGeary et al., 2019*), using concentrations of 40%, 12.65%, 4%, 1.265%, and 0.4% (v./v.) of each purified AGO–miRNA complex. For the replicate of let-7a used throughout the main text, the AGO-RBNS with 12.65% (v./v.) AGO–let-7a was performed in duplicate, and reads from both samples were considered together in all downstream analyses of this replicate.

### Analysis of *k*-mer enrichments

Positional enrichments of 8-nt *k*-mers were calculated by comparing the reads from RNA bound to 840 pM AGO2–let-7a complex to reads from the input library. For each of the two sets of reads, reads that contained one of the 18 possible 8mer-mismatch sites in the correct position (such that the CUACCUCA 8mer-consensus sequence spanned positions 26–33 of the read), but did not contain a canonical 8mer, 7mer-m8, 7mer-A1, 6mer, 6mer-m8, or 6mer-A1 site, were used to enumerate all possible 8-nt *k*-mers at each position within the library. Both count tables were normalized such that they summed to 1, and the normalized count table corresponding to the bound sample was divided by that of the input library to arrive at the enrichment of each *k*-mer at each position within the library. These *k*-mers were ranked according to the sum of the top five positional enrichments of each, considering each possible 8-nt window located entirely within the random sequence region 5′ of the programmed site. These 8-nt windows are labeled in *Figure 1D* relative to the position in the programmed site designed to pair to miRNA position 8, such that *k*-mer position 9 of *Figure 1D* corresponds to positions 18–25 when counting from the 5′ end of the random sequence region, and *k*-mer position 26 corresponds to library positions 1–8.

### Assignment of miRNA sites within reads from the programmed libraries

When counting seed sites and contiguously paired 3′ sites within the random-sequence region of the programmed libraries, reads were first queried for whether they included a canonical or 8mer-mismatch site at the programmed region (i.e., at library positions 26–33, counting from the 5′ end of the random-sequence region). Reads containing a canonical site (e.g., an 8mer or 7mer-m8 site)

in the programmed region despite its design were still counted and used for relative $K_D$ assignment, but their measured relative $K_D$ values were not used in any further analyses, owing to the ambiguity of whether the presence of these reads was the result of errors during chemical synthesis, in vitro RNA transcription, library preparation, or sequencing.

Those reads containing a seed site (defined as an 8mer, 7mer-m8, 7mer-A1, 6mer, 6mer-m8, or 6mer-A1 site, or one of the 18 possible canonical 8mer sites with a mismatch within positions 2–7) at the programmed region were further assigned to one of four categories based on whether the random region contained: (1) neither a seed site nor a 3′ site (defined as a site of 4–11 nt of contiguous complementarity to a region of the miRNA spanning position 9 to the 3′-most nucleotide), (2) at least one seed site but no 3′ sites, (3) at least one 3′ site but no seed sites, or (4) at least one seed site and at least one 3′ site. This categorization enabled read enrichment from category 3 to be compared with that of category 1 as an indication of the contribution of pairing to the 3′ end with minimal influence from seed binding outside the programmed region.

The categorization of each read proceeded as follows: (1) any canonical or 8mer-mismatch sites other than the programmed site were identified, looking within the entire random-programmed-random region with three nucleotides of constant sequence appended to the 5′ and 3′ ends of the read, (2) the 28-nt segment spanning library positions 9–37 (which included three nucleotides of the 5′ constant sequence) was queried for the longest contiguous match to the miRNA 3′ end, retaining multiple sites in the case of ties. Any site(s) 4–11 nt in length that was not contained within a canonical or 8mer-mismach site found within this 28-nt segment was counted as a 3′ site, and (3) the read was then assigned to one of the four categories described above. In the case of a read with either a single 3′ site or a single seed site within the random region, the read was assigned to that site, recording the type of site, the identity of the programmed site, and the distance between the two. In the case of a read with either multiple 3′ sites or multiple seed sites within the random region, the read count was divided equally among the sites. In the case of a read with both a 3′ site(s) and a seed site(s), the read was split between all seed-and-3′-site pairs, recording the names of the seed, 3′, and programmed site for each.

This categorization yielded tables of counts for reads with (1) only programmed sites, (2) seed-and-programmed site pairs with positional information, (3) 3′-and-programmed site pairs with positional information, (4) seed-and-3′-and-programmed site triples with no positional information, and (5) reads with neither a canonical site nor an 8mer-mismatch site at the programmed site position. These tables were either used directly for relative $K_D$ estimation, or first pooled such that the identity of their programmed site was not considered. When pooling, all counts corresponding to reads with identical site-and-positional information were summed into two categories: those for which the programmed site was an 8mer-mismatch site, and those for which the programmed site was one of the canonical sites.

## Assignment of miRNA sites within reads from the random-sequence libraries

When counting seed sites (including 8mer-mismatch sites) and contiguously paired 3′ sites within the random-sequence libraries, the 37-nt random-sequence region of each read was appended with 3 nt of constant sequence at each end, except in the case of miR-1, for which the 5′-most 36 nt of the random-sequence region was appended with 3 nt of only the 5′ constant sequence, due to the sequence bias present at the very 3′ end of these libraries caused by erroneous lack of a TCG sequence in the 3′ constant region required for pairing to the Illumina reverse-primer sequence during bridge-amplification (*McGeary et al., 2019*). The relevant portion of each read was queried for all seed sites (including 8mer-mismatch sites) and all 3′ sites between 4 and 11 nt in length, allowing individual seed sites to overlap, and individual 3′ sites to overlap. If the read contained (other than the programmed site) only seed sites or only 3′ sites, the read counts were split evenly between each site. If a read contained at least one seed site and at least one 3′ site, and if the 3′ site(s) fell upstream of a seed site, the 3′ site(s) was trimmed to remove any overlap with the seed site(s), and retained if the 3′ site(s) was ≥4 nt in length.

If one or more 3′ sites persisted for the read after accounting for any seed-site overlap, the 3′ site(s) of length equal to the longest 3′ site(s) was retained. All possible bipartite sites associated with that read were then enumerated, such that each seed site was considered to form a bipartite site with all

3′ sites that were 5′ of that seed site. The read was then split among any bipartite sites identified. In the event that no bipartite sites were identified, the read was split among all the seed and 3′ sites equally. While this procedure ensured equal partitioning of read counts in cases of multiple seed and 3′ sites within a given read, in practice only a small fraction of reads contained multiple seed sites and a 3′ site, or a seed site and multiple 3′ sites.

This categorization yielded tables of counts for reads with (1) only seed sites, (2) only 3′ sites, (3) seed-and-3′ bipartite sites with recorded inter-site spacing, and (4) neither a seed nor a 3′ site (referred to as 'no site'). These tables were either used directly for relative $K_D$ estimation, or the bipartite sites were first pooled to combine data for 3′ sites with different canonical sites or 8mer-mismatch sites prior to relative $K_D$ estimation. When pooling, all counts corresponding to reads with the same 3′ site and distance from the miRNA position 8 of the seed site were summed into two categories: those for which the seed site was an 8mer-mismatch site, and those for which the seed site was one of the canonical sites.

## Calculation of relative $K_D$ values

Relative $K_D$ values were calculated as described (*McGeary et al., 2019*).

## Correction of relative $K_D$ values using data from random-library experiments

Due to the deviation from the expected linear relationship between the relative $K_D$ values calculated for canonical sites (8mer, 7mer-m8, 7mer-A1, 6mer, 6mer-m8, and 6mer-A1 sites) and 8mer-mismatch sites (*Figure 2—figure supplement 1B*, left) we applied locally estimated scatterplot smoothing (LOESS) to generate an empirical correction to apply to these data, as has been done in analyses of metabolic-labeling data to correct mRNA abundance as a function of uridine content (*Schwanhäusser et al., 2011*). The relative $K_D$ values of the canonical sites for the programmed-library experiments used for this correction were derived from the geometric mean of the relative $K_D$ values of each site measured at each position at which they occurred within the library and in the context of each of the 18 programmed 8mer-mismatch sites, whereas the relative $K_D$ values of the 8mer-mismatch sites were derived from the reads associated with their occurrence at the programmed site in the absence of any seed sites or 3′ sites 4–11 nt in length in the random region. The relative $K_D$ values of these sites for the random-library experiments were derived from counts corresponding to instances of these sites within the reads.

We corrected the programmed-library relative $K_D$ values by calculating $R_i$, defined as:

$$R_i = \ln \frac{K_{r,i}}{K_{p,i}}, \tag{1}$$

where $K_{r,i}$ and $K_{p,i}$ refer to the relative $K_D$ values derived from the random-library and programmed-library experiments, respectively, for each site $i$. LOESS was used to fit a nonlinear function describing $R_i$ as a function of $K_{p,i}$:

$$R\left(K_p\right) \sim f_{LOESS}\left(x = \ln K_p\right). \tag{2}$$

This function was then used to correct each programmed library–derived relative $K_D$ value by multiplying each value by the output of the function with itself as input:

$$K'_{p,i} \equiv K_{p,i} \times e^{f_{LOESS}\left(x=\ln K_{p,i}\right)}. \tag{3}$$

The transformed $K'_{p,i}$ values were used throughout the study other than in *Figure 2—figure supplement 1A* and the left-hand panels of *Figure 2—figure supplement 1B*. LOESS was implemented in R using the *loess* function as part of the *stats* package, with 'span' and 'surface' parameters set to 10 and 'direct', respectively.

## Analysis of results from *Becker et al., 2019*

The 22,300 $K_D$ values measured for let-7a were inspected using the table provided as supplementary data (*Becker et al., 2019*). For each $K_D$ value, the target sequence was queried for a canonical or

8mer-mismatch site, hierarchically looking first for the 8mer, any 8mer-mismatch sites, and then any 7mer-m8, 7mer-A1, 6mer, 6mer-m8, or 6mer-A1 sites. If one or more of these sites were found, the target sequence 5′ of each site was queried for its longest stretch of complementarity to the 3′ region of let-7a (i.e., all nucleotides 3′ of position 8), and if this complementarity was between 4 and 11 nt in length, the $K_D$ value of that target sequence was assigned to that bipartite site, specifying its seed site, 3′ site, and intervening nucleotide length. If multiple 3′ sites were tied for the longest length, both 3′ sites were ascribed to the target sequence. Upon using the sequences to define the bipartite site information for each target RNA, only those target RNAs with a single bipartite site, or a single seed site and no 3′ pairing between 4 and 11 nt in length, were included in the downstream analyses. From these sites, we calculated the $K_D$ fold change for each available 3′-site, offset, and seed-site combination (*Figure 2—figure supplement 2*) by dividing the geometric mean of the $K_D$ values of target RNAs containing that bipartite site by the geometric mean of that of the target RNAs containing the seed site with no 3′ pairing, except when calculating the $K_D$ fold-change values for bipartite sites with the 8mer-xA5 seed site (*Figure 2—figure supplement 2F*). Because no target RNAs fit our criteria as containing only the 8mer-xA5 site and no 3′ pairing between 4 and 11 nt in length, we used 10 nM as the reference $K_D$, which was the lower limit of detection measured, and was the measured $K_D$ for seven of the 16 8mer-mismatch sites present in the data.

## let-7a reporter library

A reporter-plasmid library was designed to assay the intracellular efficacies of 15 synthetic let-7a 3′-compensatory sites (*Figure 3A*). These sites were designed to test the interplay between the effects of the position, length, and offset of 3′ pairing to let-7a. Each of the 15 3′-compensatory sites had an 8mer-w6 match to the seed region (i.e., an 8mer match to the seed region disrupted by a wobble at position 6), 3′-pairing spanning one of three ranges (matching either positions 13–16, positions 11–19, or positions 13–21 of let-7a), and either no nucleotides, A, AAAA, U, or UUUU inserted between the target nucleotides opposite let-7a positions 9 and 10 (causing the target segment linking the seed and 3′ complementary to impart an offset of 0, +1, +4, +1, or +4 nt, respectively). For comparison, reporters were also designed to assay the efficacy of five sites with only seed pairing (the canonical 6mer, 7mer-A1, 7mer-m8, and 8mer sites, plus the 8mer-w6 site lacking compensatory pairing) and seven no-site sequences (sites that had no more than five contiguous pairs to let-7) (*Figure 3A*). Reporters were also designed to assay efficacy of the dual 3′-compensatory sites that mediate *lin-41* repression in *C. elegans* (*Figure 3A*).

The placement and spacing of the sites were based on the placement and spacing of the dual 3′-compensatory sites within the 3′ UTR of *C. elegans lin-41*, wherein position 1 of each site is separated by 46 nt (after discounting the +1-nt offset nucleotide of the downstream site). For example, in the dual-site configuration of the *lin-41* sequence context, the site under investigation replaced each of the two 3′-compensatory sites within the segment spanning nucleotides 650–798 of the *lin-41* 3′ UTR. Likewise, in each of the single-site configurations, the site replaced one of the two endogenous sites, whereas a no-site sequence replaced the other endogenous site. When in the other 13 de novo sequence contexts, sites were located at the same position as they were in the *lin-41* sequence context, but the *lin-41* sequence was replaced with arbitrary sequence of equal length. Each of the arbitrary sequence segments, as well as the no-site sequence segments, was generated by weighted sampling of dinucleotides according to the frequencies observed human 3′-UTR sequences, except at positions opposite let-7a nucleotides 9 and 10, where only the dinucleotides that would place an A or U at these two positions were used. For the *lin-41* 3′-UTR context, each of the central U nucleotides within the linking segments of the upstream (GUU) and downstream (AUU) sites was either removed to generate the 0-offset possibility, or replaced to generate the +1 and +4 possibilities, and the 9-nt 8mer-bA5 of the upstream site was replaced with 8 nt of sequence. The de novo 3′-UTR contexts and the no-site replacement sequences were also designed to exclude any (1) ≥6-nt contiguous match to the first eight positions of miR-1, (2) ≥6-nt contiguous match to the first 11 positions of let-7a, (3) ≥4-nt contiguous match to positions 9–21 of let-7a, (4) PUMILIO response element (UGUANAUA), (5) splice-donor sequence (GGGURAGU), (6) BstXI restriction site (CCAN₆UGG), (7) BsrGI restriction site (UGUACA), or (8) poly(A) signal (AAUAAA). In total 952 variants were designed [(((15 3′-compensatory sites + 5 sites with only seed pairing) × (three configurations)) + (7 no sites + 1 *lin-14* site)) × 14 contexts = 952 variants].

The sequences of these 952 variants were inserted into an expression vector (*McGeary et al., 2019*) at a position corresponding to nucleotide 36 of the reporter transcript 3′ UTR. The vector drives expression of an EGFP mRNA from the pEF1a promoter, with a 939-nt EF-1α intron 49 nt upstream of the coding sequence. The length of the inserted sequence ranged from 146 bp (for inserts with no sites, seed-only sites, or 3′-compensatory sites with 0-nt offsets) to 154 bp (for inserts with dual 3′-compensatory sites with 4-nt offsets). The DNA library of inserts was synthesized (Twist Biosciences, Oligo Pools order, *Supplementary file 1*) and amplified (KAPA HiFi HotStart DNA Polymerase, Roche Diagnostics, 07958935001), using primers that added a BsrGI site on one end of the amplicon, a BstXI site on the other end of the amplicon, and homology to a PCR primer used to prepare directed RNA-seq libraries (*Supplementary file 2*). After double-digest with BstXI and BsrGI, the amplicon was incubated with the large fragment from a BstXI and BsrGI double-digest of the parental plasmid in a T4 DNA Ligase reaction (New England Biolabs, M0202S). Following ethanol precipitation, the ligation mix was electroporated into OneShot Top10 Electrocomp *E. coli* (Thermo Fisher, C404050) in two separate reactions with 100 µL of cells each, and bacteria from both electroporations were incubated in SOC media for 90 min at 37°C to recover, and then plated onto 19 10-cm LB agar plates containing ampicillin. In parallel, a small fraction of the cells were dilution-plated, which indicated that approximately five million independent transformants were obtained. After 16 hr of bacterial growth under ampicillin selection at 37°C, bacteria were harvested, and the reporter-plasmid library was purified by MAXI-prep (Qiagen, 12362).

## Massively parallel reporter assay

The assay was performed in duplicate for each of the four conditions (three miRNA duplex transfections and a mock miRNA transfection), generating eight samples. For each condition, F9 cells were plated at 3 million cells in one 10-cm dish supplied with 10 mL media (DMEM +10% FBS). After 24 hr of culture, the cells were supplied with fresh media and were transfected with 5.8 µg of the plasmid library diluted with 28.9 µg of pUC19 carrier plasmid using Lipofectamine 2000 (Thermo Fisher, 11668019) and Opti-MEM (Thermo Fisher 31985062), and either let-7a duplex, let-7a-21nt duplex, miR-1 duplex, or no duplex (mock) at a final duplex concentration of 25 nM, according to the manufacturer's protocol. After 24 hr, cells lysis, RNA extraction, reverse transcription, PCR, and gel purification were performed to generate amplicons for RNA-seq, as described (*McGeary et al., 2019*), with the following modifications: (1) for the second replicate, no SUPERase•In was added to the lysis buffer, (2) one-fifth of the recovered RNA from each sample, rather than one-half, was treated with TURBO DNase and used in the downstream reverse transcription reaction, (3) for the final PCR reaction, 30 µL of cDNA was amplified in three 100-µL PCR reactions, rather than 56 µL of cDNA in seven 100 µL PCR reactions, (4) after the final PCR step, the three 100-µL PCR reactions were pooled, and one third of the material was gel purified, rather than pooling, EtOH-precipitating, resuspending, and gel-purifying all of the material, and (5) the material was purified by Metaphor agarose–gel electrophoresis (Lonza Bioscience, 50180), rather than formamide-gel electrophoresis. The eight samples were sequenced with multiplexing on two lanes of an Illumina HiSeq 2500 run in rapid mode with 120-nt single-end reads.

For analysis, reads were first filtered for quality, as described (*McGeary et al., 2019*). Reads passing these criteria were assigned to one of the 952 sequences of the library, requiring a perfect match to the sequence. For each sequence, counts were normalized to the total number of perfectly matching counts to obtain counts per million (cpm). Fold-change values reported for each sample in the top panels of *Figure 3B* and *Figure 3—figure supplements 1 and 2*, were calculated by determining, for each of the 20 sites $i$ and two positions $j$, $r_{i,j}$, given by:

$$r_{i,j} = \log_2 \left( \frac{y_{i,j,+}/y_{0,+}}{y_{i,j,-}/y_{0,-}} \right), \tag{4}$$

where $y_{i,j,+}$ refers to the summed cpm values over all 14 site contexts for either the upstream-only ($j = u$) or downstream-only ($j = d$) site configuration for a library from a miRNA-transfected sample, $y_{i,j,-}$ refers to the same value but for the mock-transfected sample, $y_{0,+}$ refers to the summed cpm values over all $7 \times 14$ dual no-site reporters for the miRNA-transfected sample, and $y_{0,-}$ refers to the same value but for the mock-transfected sample. Because the experiment was performed with two different 3′ isoforms of let-7a (i.e., the 22-nt let-7a and the 1-nt shorter isoform, let-7a-21nt), and

because results for these two let-7a isoforms were indistinguishable, these results were treated as replicates. This led to eight $r_{i,j}$ values for each site in the let-7a arm of the experiment (two single-site positions × 2 miRNA isoforms × 2 biological replicates), and four $r_{i,j}$ values for each site in the miR-1 control arm. Fold-change values reported in the bottom panels of *Figure 3B* and *Figure 3—figure supplement 1* were similarly determined by summing cpm values over all 14 site contexts for the dual-site configuration ($j = b$) (except when measuring the efficacy *lin-41* sites only when within the *lin-41* 3'-UTR context). This yielded four values for each site in the let-7a arm and two for each site in the miR-1 control arm. The p values testing the significance of differences in mean repression were generated by performing Tukey's HSD test to calculate the significance of all pairwise differences between the means of the 42 categories shown in *Figure 3B* (with the no-site category counted once), using the TukeyHSD function in R.

The model relating the measured relative $K_D$ of each site possibility $i$, $K_i$ to its predicted efficacy of repression $r_i$ in *Figure 3C* was given by:

$$r_i = \log_2 \left( \frac{1}{1 + bN_i} \right),$$
(5)

where

$$N_i = \frac{a}{a + K_i},$$
(6)

with $a$ specifying the concentration the free AGO–miRNA complex in the cytoplasm of the F9 cells, and $b$ specifying the ratio of degradation of fully bound versus unbound mRNA subtracted by one. These two parameters were fit to maximize the agreement of the predicted fold-change values with the measured fold-change values, with final values of $a = 2.79 \times 10^{-3}$ and $b = 8.49 \times 10^{-1}$. Additionally, relative $K_D$ values for 3'-compensatory sites with pairing to positions 13–16 and nonzero offsets or for sites with 9 nt of 3' pairing and a +1-nt offset were calculated using only reads compatible with that site (e.g., ACAANNN<u>AAAA</u>NCUGCCUCA for the 4mer-m13–16, +4$_A$-nt possibility; and UAUA-CAACCNUNCUGCCUCA for the 9mer-m11–19, +1$_U$-nt possibility). However, for sites with 9 nt of 3' pairing and +4-nt offsets, read numbers were not sufficient for this approach, and relative $K_D$ values were calculated using reads with either a W (A or U) at each of the insertion positions, with the +4-nt offset 9mer-m11–19 possibilities further corrected by first calculating, for each of the +4$_A$-nt and +4$_U$-nt possibilities, the sum of the fractional abundance of that site across all five binding samples (when averaging the two 12.65%-binding samples) divided by the number of reads in the input sequencing library with either AAAA or UUUU within the 25-nt random region, and then dividing and multiplying respectively the relative $K_D$ values of the +4$_A$-nt and +4$_U$-nt possibilities by the square root of the ratio of the two values. The +4-nt offset 9mer-m13–21 possibilities could not be corrected in this way because of the lack of any reads with these sites within the bound samples.

## Calculation of pairing and offset coefficients

In order to separate the intrinsic pairing preferences of each miRNA 3' end from the effects of varying the offset of pairing, we fit a thermodynamic model of 3'-end binding efficacy to the $K_D$ fold-change values measured when summing the counts from each of the 18 8mer-mismatch sites. The model was constructed to produce a $\log_{10}$-tranformed $K_D$ fold-change value, denoted here using $\kappa$, as a function of the 5' terminus of the pairing $i$, the 3' terminus of pairing $j$, and the offset between the seed and 3' pairing $k$. In order to make no assumptions regarding the thermodynamic nature of 3'-end binding (e.g. that each nucleotide would contribute independently to the binding energy), and as well to make no assumptions about the nature of the offset preferences, the model included two sets of categorical coefficients, one set $\alpha_{i,j}$ describing the 3'-pairing range as a function of the 5' and 3' termini of pairing indices $i$ and $j$, and another set $\beta_k$ describing the offset preferences as a function of the offset index $k$:

$$\kappa(i, j, k) = \kappa(\alpha_{i,j}, \beta_k).$$
(7)

Because the nature of the relationship between the pairing range and the offset preferences could not be known a priori, we constructed three variants of the model function $\kappa(i, j, k)$:

$$\kappa_a\left(i,j,k\right) = \alpha_{i,j} + \beta_k$$
$$\kappa_m\left(i,j,k\right) = \alpha_{i,j}\beta_k \tag{8}$$
$$\kappa_{mc}\left(i,j,k\right) = \alpha_{i,j}\beta_k + \gamma,$$

where $\kappa_a\left(i,j,k\right)$, $\kappa_m\left(i,j,k\right)$, and $\kappa_{mc}\left(i,j,k\right)$ describe additive, multiplicative, and multiplicative-plus-constant models. We note that an additive-plus-constant variant is trivially equivalent to an additive model, since the constant term can be subsumed by either of the $\alpha_{i,j}$ or $\beta_k$ coefficients.

Each of the models described by *Equation 8* were fit to the data by minimizing a cost function giving the sum of the squared difference between each of the measured $\log_{10}$-transformed $K_D$ fold-change values $y_{i,j,k}$ and their corresponding model predictions $\kappa\left(i,j,k\right)$ for all pairing-range and offset combinations $i$, $j$, and $k$ with 3′-pairing lengths 4–11 bp and offset between −4 and +16 nt:

$$f_{cost,\kappa}\left(\boldsymbol{\alpha},\boldsymbol{\beta}\right) = \sum_{i=9}^{n_m-3} \sum_{j=i+3}^{n_m} \sum_{k=-4}^{+16} \left(y_{i,j,k} - \kappa\left(i,j,k\right)\right)^2, \tag{9}$$

where $\boldsymbol{\alpha}$ and $\boldsymbol{\beta}$ represent the vector of all $\alpha_{i,j}$ and all $\beta_k$ coefficients, respectively, $\gamma$ represents a fitted constant, and $n_m$ represents the length of the miRNA. This cost function was minimized with the *optim* function in R using the L-BFGS-B method, supplying the cost function and its gradient and setting the 'maxit' parameter to $1 \times 10^7$. When optimizing all three models, all $\boldsymbol{\alpha}$, $\boldsymbol{\beta}$, and $\gamma$ parameters were initialized at 1.0 and bounded between 0.0 and 10.0 during the optimization.

Because the multiplicative model $\kappa_m\left(i,j,k\right)$ ($r^2$ = 0.92, 0.86, and 0.96 for let-7a, miR-1, and miR-155, respectively, *Figure 5—figure supplement 1D*) performed significantly better than the additive model $\kappa_a\left(i,j,k\right)$ ($r^2$ = 0.81, 0.81, and 0.94), and because the multiplicative-plus-constant model $\kappa_{mc}\left(i,j,k\right)$ provided only marginally increased performance ($r^2$ = 0.93, 0.86, and 0.96) while decreasing model interpretability (as the constant term corresponded to a benefit to binding irrespective of the manner of the pairing to the miRNA 3′ end), we selected the multiplicative model. For the purposes of interpretation of the model coefficients, we re-scaled the coefficients as follows:

$$\boldsymbol{\alpha}' = \boldsymbol{\alpha} \times \max\boldsymbol{\beta}$$
$$\boldsymbol{\beta}' = \frac{\boldsymbol{\beta}}{\max\boldsymbol{\beta}}, \tag{10}$$

which, because none of the coefficients were negative, caused each offset coefficient $\beta'_k$ to fall between 0.0 and 1.0, thereby corresponding to a different fractional reduction in binding energy for each offset $k$. Each re-scaled pairing range coefficient $\alpha'_{i,j}$ therefore also represented the maximum $K_D$ fold change that could be obtained by contiguous pairing to nucleotides $i$ through $j$.

We estimated the model error by calculating the asymptotic covariance matrix $\boldsymbol{V}\langle\boldsymbol{\theta}\rangle$ using the standard approximation

$$\boldsymbol{V}\langle\hat{\boldsymbol{\theta}}\rangle = \frac{f_{cost,k}(\hat{\boldsymbol{\theta}})}{n-p}\left(\boldsymbol{A}\langle\hat{\boldsymbol{\theta}}\rangle^T\boldsymbol{A}\langle\hat{\boldsymbol{\theta}}\rangle\right)^{-1}, \tag{11}$$

where $\hat{\boldsymbol{\theta}}$ is the vector of all optimal pairing and offset coefficients, $n$ is the total number of data points, $p$ is the total number of model parameters (i.e., the length of vector $\hat{\boldsymbol{\theta}}$), and $\boldsymbol{A}\langle\hat{\boldsymbol{\theta}}\rangle$ represents the matrix of partial derivatives $\frac{\partial\kappa}{\partial\theta}$ (*Seber and Wild, 2003*). From the covariance matrix, the 95% confidence intervals for each model coefficient are given by

$$\hat{\boldsymbol{\theta}} \pm t_{\alpha=0.975,\nu=n-p}\sqrt{\mathrm{diag}\left(V\langle\hat{\boldsymbol{\theta}}\rangle\right)}, \tag{12}$$

where $t_{\alpha=0.975,\nu=n-p}$ represents the $t$ statistic for 97.5% confidence with $n-p$ degrees of freedom. Because the form of the model described allows the cost function to be minimized with an infinite number of distinct solutions (where, given a particular optimal $\hat{\boldsymbol{\theta}}$ comprised of $\boldsymbol{\alpha}$ and $\boldsymbol{\beta}$, any $\hat{\boldsymbol{\theta}}'$ comprised of $c\hat{\boldsymbol{\alpha}}$ and $\hat{\boldsymbol{\beta}}/c$ is an equivalent solution), the matrix given by $\boldsymbol{A}\langle\hat{\boldsymbol{\theta}}\rangle^T\boldsymbol{A}\langle\hat{\boldsymbol{\theta}}\rangle$ is not linearly independent, and thus cannot be inverted as required in *Equation 11*. This issue can be circumvented by arbitrarily fixing one parameter in the course of the optimization. We therefore optimized the model 21 times, fixing each $\beta_k$ coefficient at 1.0 during the optimization, determining the 95% confidence intervals for all other coefficients, and then rescaling all parameters. This led to 21 different estimates

of the confidence intervals for each pairing coefficient, and 20 distinct estimates of the confidence intervals for each offset coefficient, which were averaged to produce the error estimates reported throughout the study. We note that because the parameters were re-scaled after both the optimization and confidence interval calculation, the final, re-scaled parameter values obtained were identical in each of the 21 optimization routines.

## Theoretical relationship between $K_D$ fold change and $\Delta G$ of 3′ pairing

Because we were interested in the expected relationship between the measured $K_D$ fold-change values (and, by extension, the model-determined pairing coefficients), we derived their theoretical relationship by comparison of the apparent $K_D$ for a biochemical model in which only seed pairing was allowed (model A) with that in which both seed and 3′ pairing were allowed (model B). Model A is described by

$$R + A \overset{K_1}{\leftrightarrow} C_1, \tag{13}$$

where $R$ is the target RNA, $A$ is the unbound AGO–miRNA complex, $C_1$ is the AGO–miRNA complex seed pairing to the target RNA, and $K_1$ is the equilibrium constant describing the forward reaction, defined by

$$K_1 = \frac{[C_1]}{[R]\,[A]}, \tag{14}$$

where $[N]$ denotes the concentration of each species $N$. The fractional occupancy of target RNA $R$ with model A, $\theta_A$ , is given by

$$\theta_A = \frac{[C_1]}{[R] + [C_1]}, \tag{15}$$

and when substituted for $[C_1]$ using **Equation (14)** yields the familiar equation

$$\theta_A = \frac{[A]}{1/K_1 + [A]}, \tag{16}$$

in which $K_1$ is the reciprocal of $K_D$.

Model B is described by

$$R + A \overset{K_1}{\leftrightarrow} C_1 \overset{K_2}{\leftrightarrow} C_2, \tag{17}$$

where, in addition to the terms defined for model A, $C_2$ is the AGO–miRNA complex bound to the target RNA with both seed and 3′ pairing, and $K_2$ is the equilibrium constant describing the forward reaction, defined by

$$K_2 = \frac{[C_2]}{[C_1]}, \tag{18}$$

which mechanistically corresponds to the AGO–miRNA complex engaging in 3′ pairing with a target to which it is already seed-paired. We note that this model does not allow formation of $C_2$ through an intermediate in which 3′ pairing occurs prior to seed pairing. Such a model was not used because 3′-autonomous pairing is considerably rarer than seed pairing and because it prevented the derivation from becoming unnecessarily complex.

The fractional occupancy of the target RNA in model B is given by

$$\theta_B = \frac{[C_1] + [C_2]}{[R] + [C_1] + [C_2]}, \tag{19}$$

which can be substituted for $C_2$ using **Equation (18)** to yield

$$\theta_B = \frac{[C_1]\,(K_2 + 1)}{[R] + [C_1]\,(K_2 + 1)}, \tag{20}$$

and then further substituted for $C_1$ using *Equation (14)* to yield

$$\theta_B = \frac{K_1 \, [R] \, [A] \, (K_2 + 1)}{[R] + K_1 \, [R] \, [A] \, (K_2 + 1)}.$$

(21)

This can be simplified and rearranged in an analogous form to *Equation (16)*:

$$\theta_B = \frac{[A]}{\frac{1}{K_1 (K_2 + 1)} + [A]},$$

(22)

from which it is clear that the apparent $K_D$ value is $\frac{1}{K_1 (K_2+1)}$. This indicates that the expected (beneficial) fold change to the $K_D$ is $K_2 + 1$, and that the $K_2$, rather than the $K_D$ fold change itself, would be expected to relate to the $\Delta G$ through the equation $K_2 = e^{-\Delta G/RT}$. We explored incorporating this approach by fitting the model described in the prior section to $\log_{10}[(K_D \text{ fold change}) - 1]$ instead of $\log_{10}(K_D \text{ fold change})$, but ultimately chose not to include this conversion in our analyses, because it caused many of the lowest-affinity pairing coefficients to be fit to anomalously large negative values. This occurred because those pairing coefficients were fit using $K_D$ fold-change values that in some cases closely approached 1 (i.e. the 3′ pairing provided almost no benefit), which is a regime in which small changes in measured $K_D$ fold change will lead to much larger changes in the modeled pairing coefficient. For example, a $K_D$ fold change of 1.1 would correspond to an apparent pairing coefficient of $\log_{10}(1.1–1) = −1$, whereas 1.01 would correspond to an apparent pairing coefficient of $\log_{10}(1.01–1) = −2$, or 10-fold worse binding.

## Nearest neighbor model–based prediction of 3′-compensatory pairing ΔG

For comparison with each pairing-range coefficient beginning at position $i$ and ending at position $j$, the predicted $\Delta G$ of duplex formation between the sequence of the miRNA beginning at position 9 and the sequence reverse-complementary to miRNA positions $i$–$j$, with no non-complementary nucleotides appended to either terminus, was calculated using RNAduplex, as part of the ViennaRNA package, through its Python interface (*Lorenz et al., 2011*).

## Calculation of mismatch coefficients

We extended the thermodynamic model of 3′-end binding efficacy to include seed-mismatch effects, by using as input data the $\log_{10}(K_D \text{ fold-change})$ values measured for each of the 18 8mer-mismatch sites separately. This model took the form of:

$$\kappa_2 (i, j, k, l) = \alpha_{i,j} \beta_k \delta_l.$$

(23)

where $\alpha_{i,j}$ and $\beta_k$ represented the pairing and offset preferences as before, and $\delta_l$ represented the additional set of 18 seed-mismatch coefficients. The updated cost function was therefore

$$f_{cost,\kappa_2} (\boldsymbol{\alpha}, \boldsymbol{\beta}, \boldsymbol{\delta}) = \sum_{i=9}^{n_m-3} \sum_{j=i+3}^{n_m} \sum_{k=-4}^{+16} \sum_{l=1}^{18} \left( y_{i,j,k,l} - \kappa_2 (i,j,k,l) \right)^2,$$

(24)

where $\boldsymbol{\delta}$ represents the vector of all $\delta_l$ coefficients. The optimization was performed identically as before, with the $\delta$ parameters initialized at 1.0, and bounded between 0.0 and 10.0 during the optimization. After the optimization, the coefficients were re-scaled as

$$\begin{aligned}
\boldsymbol{\alpha}' &= \boldsymbol{\alpha} \times \max \boldsymbol{\beta} \times \text{mean } \boldsymbol{\delta} \\
\boldsymbol{\beta}' &= \frac{\boldsymbol{\beta}}{\max \boldsymbol{\beta}} \\
\boldsymbol{\delta}' &= \frac{\boldsymbol{\delta}}{\text{mean } \boldsymbol{\delta}},
\end{aligned}$$

(25)

which preserved the same interpretation of each $\alpha'_{i,j}$ and $\beta'_k$ coefficient as for the prior model, and further parameterized each $\delta'_l$ to represent the multiplicative deviation in binding caused by each seed-mismatch type $l$, such that the average effect of all 18 mismatches was that of being multiplied by 1.

The 95% confidence intervals of this model also used *Equations (11) and (12)*. However, because this model was the product of three sets of categorical coefficients, one coefficient from each of two sets was required to be fixed while performing the error determination. We therefore optimized the model 21 × 18 times, fixing one $\beta_k$ coefficient at 1.0 and one $\delta_k$ at 1.0 during the optimization, determining the 95% confidence intervals for all other coefficients, and then rescaling all parameters. This led to 21 × 17 = 357 different estimates of the confidence intervals for each seed-mismatch coefficient, which were averaged to produce the error estimates reported throughout the study.

## Empirical assessment of the influence of seed-type on 3′ pairing with the random-library AGO-RBNS experiments

When analyzing the effects of seed type on $K_D$ fold change in the random-library experiments, we first attempted to apply the modeling approach as used when analyzing the data from the programmed-library experiments. However, we found that the read counts assigned to each possible pairing, offset, and seed-mismatch combination were too sparse, such that modeling using these data could not be reliably performed. We therefore analyzed the differences in the benefit of 3′-supplementary pairing between each of the six canonical sites (8mer, 7mer-m8, 7mer-A1, 6mer, 6mer-m8, and 6mer-A1) as well as a representative 3′-compensatory site given by summing the read counts of all 18 8mer-mismatch sites.

To compare these sites, we first took, for each miRNA, all 3′ pairing possibilities with lengths of 4 or 5 bp and pairing to miRNA positions 9–18, and determined for each of these possibilities the offset that imparted the maximum average $\log_{10}(K_D$ fold change), thereby constructing a 7 (site types, i.e., the six canonical sites and a combination of the 18 8mer-mismatch sites) × 20 (pairing range) matrix of $\log_{10}(K_D$ fold change) values. This matrix was then sorted by the average $\log_{10}(K_D$ fold change) of each pairing range, and then this value was subtracted from each column, such that the values within each column reported on the deviation of each site type from the average, for that pairing-and-offset possibility. These deviations were then averaged for each of the seven site types over the top five (i.e. the top quartile) of pairing-and-offset possibilities, to give the empirical contribution of each site type to 3′-binding affinity, in comparison to that of the average. These values are plotted for all six miRNAs in *Figure 5H*, and the data tables from which they were calculated are visualized in *Figure 5—figure supplement 8*, with the columns used as input for the final averaging indicated.

## Assignment of miRNA sites with imperfect 3′ pairing within reads from the programmed libraries

To calculate the effect of a mismatch, bulge, or deletion on the binding affinity of a 3′ site, the site counting was repeated for each fully paired 3′ site while also counting each of its derivatives containing either a single mismatched, bulged, or deleted nucleotide (hereafter referred to as 'imperfect 3′ sites'). This was done to reduce the total number of sites being simultaneously counted and used to calculate relative $K_D$ values, and as well to reduce assignment problems owing to any imperfect 3′ site from one region of the miRNA being sequence-identical to that from any other region of the miRNA.

The site counting was performed similarly to that of the fully paired 3′ sites, with some differences: those reads containing a seed site at the programmed region were still assigned to one of four categories, with the definition of a 3′ site expanded to include any fully paired 3′ site of length 4–11 nt in addition to any imperfect derivatives of that site. The categorization of each read proceeded through the following steps: (1) Any seed sites were identified, looking within the entire random-programmed-random region with 3 nt of constant sequence appended to the 5′ and 3′ ends of the read. (2) The 28-nt segment comprising 3 nt of the 5′-constant sequence and 25 nt of random sequence was queried for any instances of any imperfect 3′ sites, with any deleted- or mismatched-nucleotide sites that were contained with another mismatched- or bulged-nucleotide site not counted (e.g., the let-7a 11mer-m11–20 with a mismatched U at position 20 is inherently contained within the 11-mer-m11–20 with a bulged U between positions 19 and 20, but only the bulged-nucleotide version of the site would be recorded). Any imperfect 3′ sites were also queried to make sure that the nucleotides on each side of the site were not complementary to the next corresponding positions of the miRNA guide. We note that if such an imperfect 3′ site was found that failed these criteria, any identified fully paired 3′ sites were not counted toward that read. (3) The 28-nt segment comprising 3 nt of the 5′-constant sequence and 25 nt of random sequence was queried for the longest fully paired 3′ site(s), retaining

multiple 3′ sites if more than one was of the longest length. If at least one fully paired 3′ site >4 nt in length and not contained within a seed site (if such sites were present) was identified, and if at least one imperfect 3′ sites was identified, the length of the fully paired 3′ site(s) was compared to that of the imperfect 3′ site(s), and the 3′ site(s) that was longer was retained. Lastly, if the fully paired 3′ site(s) was ≥11 nt in length, neither the fully paired nor imperfect 3′ site(s) were counted. (4) The read was then assigned one of the four categories, and the read count split as described when assigning contiguous 3′ sites.

The relative $K_D$ values used in *Figure 7* and *Figure 7—figure supplement 1* were calculated using the summed counts of each site over all 18 mismatch sites in the programmed region. In order to mitigate noise in the measurement of affinities for individual imperfect 3′ sites, the $K_D$ fold-change values of each fully paired site and each of its imperfect sites at particular offset values were substituted with the geometric mean of the $K_D$ fold change of those sites over the three contiguous offset values centered on that offset value. The offset value used for reporting of the imperfect $K_D$ fold-change values was the offset value between −2 and +6 nt for which the $K_D$ fold-change of the fully paired site was the greatest.

When performing the positional analyses of *Figure 7E*, ΔΔG values corresponding to sites in which the position of a particular bulged nucleotide was ambiguous were used to calculate the average ΔΔG value for each of the inter-nucleotide positions at which it might occur. Conversely, these sites were omitted from comparisons of terminal bulges and mismatches of *Figure 7F*.

## Assignment of miRNA sites with imperfect 3′ pairing within reads from the random-sequence libraries

When counting imperfect 3′ sites within the random-library reads, we similarly preformed the read counting and relative $K_D$ fitting for each fully paired site and its imperfect derivatives. Individual reads were queried for all seed sites and fully paired 3′ sites as described for the fully paired site counting with the random-library experiments, in addition to being queried for any imperfect 3′ sites derived from a fully paired site. If a read contained at least one seed site and at least one fully paired or imperfect 3′ site, each 3′ site was checked for any amount of overlap with any seed sites. If a fully paired 3′ site overlapped a seed site with the 3′ site being the 5′-most site, the 3′ site was trimmed to exclude the region overlapping the seed site, and retained if the site was still ≥4 nt in length. As before, any fully paired 3′ sites that either overlapped any seed sites from the 3′ end, contained a seed site within them, or were entirely contained within a seed site, were discarded. Any imperfect 3′ sites identified were examined to ensure that read positions just outside their limits were not complementary to the corresponding miRNA positions, and as well that they did not overlap any of the seed sites within the read.

If one or more 3′ site (either fully paired or imperfect) remained, all of the fully paired 3′ sites of length equal to that of the longest fully paired 3′ site in the read were retained. If at least one fully paired and at least one imperfect 3′ site remained, the length of the fully paired 3′ site(s) was compared to that of the fully paired site from which the imperfect site(s) was derived, and if it was shorter, the imperfect 3′ site(s) was retained, and the fully paired site(s) was discarded. If the fully paired 3′ site(s) was longer than the fully paired site from which the imperfect site(s) was derived, the fully paired site(s) was retained, and the imperfect site(s) was discarded. All possible bipartite sites associated with that read were then enumerated, whereby each seed site was considered to form a bipartite site with each 3′ sites that was fully 5′ of that seed site. The read was then split among any bipartite sites identified. In the event that no bipartite sites were identified, the read was split among all the seed and 3′ sites equally. While this procedure ensured equal partitioning of read counts in the case of multiple seed or 3′ sites within a read, in practice only a small fraction of reads contained either multiple seed sites and a 3′ site or a seed site and multiple 3′ sites.

The $K_D$ fold-change values associated with each imperfect site were calculated from the ratio of the relative $K_D$ value of the site when summing the counts for all 18 8mer-mismatch sites to the geometric mean of the relative $K_D$ values of each of the 18 8mer mismatch sites calculated individually. As with the relative $K_D$ values corresponding to the imperfect sites derived from the programmed libraries, the values reported for each of the perfect and imperfect sites reported in *Figure 7—figure supplement 2* were the geometric mean of the values from the three contiguous offsets at which the $K_D$ fold change of the fully paired site was the greatest.

## Acknowledgements

We thank K Lin for helpful discussions, the Whitehead Genome Technology Core for high-throughput sequencing, and members of the Bartel lab for comments on this manuscript.

## Additional information

### Funding

| Funder | Grant reference number | Author |
|---|---|---|
| National Institute of General Medical Sciences | GM118135 | David P Bartel |
| National Institute of General Medical Sciences | GM123719 | Namita Bisaria |
| Howard Hughes Medical Institute | | David P Bartel |

The funders had no role in study design, data collection and interpretation, or the decision to submit the work for publication.

### Author contributions

Sean E McGeary, Conceptualization, Data curation, Formal analysis, Investigation, Methodology, Visualization, Writing - original draft, Writing – review and editing; Namita Bisaria, Conceptualization, Data curation, Formal analysis, Funding acquisition, Investigation, Methodology, Visualization, Writing - original draft, Writing – review and editing; Thy M Pham, Peter Y Wang, Investigation, Writing – review and editing; David P Bartel, Conceptualization, Funding acquisition, Project administration, Resources, Supervision, Writing - original draft, Writing – review and editing

### Author ORCIDs

Sean E McGeary (ID) http://orcid.org/0000-0001-5343-6447
Namita Bisaria (ID) http://orcid.org/0000-0003-3641-5438
Thy M Pham (ID) http://orcid.org/0000-0001-5281-9147
Peter Y Wang (ID) http://orcid.org/0000-0002-6129-2815
David P Bartel (ID) http://orcid.org/0000-0002-3872-2856

### Decision letter and Author response

Decision letter https://doi.org/10.7554/eLife.69803.sa1
Author response https://doi.org/10.7554/eLife.69803.sa2

## Additional files

### Supplementary files

- Supplementary file 1. Library of DNA sequences used for let-7a massively parallel reporter assay.
- Supplementary file 2. Oligonucleotides used in this study.
- Transparent reporting form

### Data availability

Sequencing data have been deposited in GEO; accession GSE196458.

The following dataset was generated:

| Author(s) | Year | Dataset title | Dataset URL | Database and Identifier |
|---|---|---|---|---|
| McGeary SE, Bisaria N, Pham TM, Wang PY, Bartel DP | 2022 | MicroRNA 3'-compensatory pairing occurs through two binding modes, with affinity shaped by nucleotide identity and position | https://www.ncbi.nlm.nih.gov/geo/query/acc.cgi?acc=GSE196458 | NCBI Gene Expression Omnibus, GSE196458 |

The following previously published dataset was used:

| Author(s) | Year | Dataset title | Dataset URL | Database and Identifier |
|---|---|---|---|---|
| McGeary SE, Lin KS, Shi CY, Pham TM, Bisaria N, Bartel DP | 2019 | The biochemical basis of microRNA targeting efficacy | https://www.ncbi.nlm.nih.gov/geo/query/acc.cgi?acc=GSE140220 | NCBI Gene Expression Omnibus, GSE140220 |

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
