## [Editor Report]

This manuscript will be of interest to readers in the field of microRNA (miRNA) biology, particularly those interested in miRNA targeting. The authors interrogated non-canonical miRNA target recognition to a depth vastly exceeding any study to date. The results revealed unexpected, sequence-specific diversity in miRNA-targeting modes, providing new insights relevant for improved target prediction.

---

## [Decision Letter]

**Decision letter after peer review:**

Thank you for submitting your article "Pairing to the microRNA 3′ region occurs through two binding modes, with affinity shaped by nucleotide identity and position" for consideration by *eLife*. Your article has been reviewed by 3 peer reviewers, and the evaluation has been overseen by a Reviewing Editor and James Manley as the Senior Editor. The following individuals involved in review of your submission have agreed to reveal their identity: Andrea Ventura (Reviewer #2); Javier Martinez (Reviewer #3).

Essential revisions:

All three reviewers and the reviewing editor found this to be a strong and significant manuscript that addresses important questions relevant to miRNA targeting. Nevertheless, each reviewer has made a few comments that if addressed would improve the manuscript. Specifically, it was agreed that some reporter assays as mentioned in two of the reviews would be important. Secondly, it was also agreed that the paper was quite dense and in some places difficult to read. Please consider this aspect during revision. It is important that the work be comprehensible by the general reader of *eLife* and not just those who appreciate all the details.

*Reviewer #1 (Recommendations for the authors):*

1. The study is an impressive computational feat with many interesting discoveries. However, the current manuscript is so long and time-consuming to absorb that the majority of the broader audience may lack the will to invest in reading the work and appreciating its full implications. We therefore recommend the authors shorten the manuscript with an emphasis on distilling and crystallizing the most salient points.

2. Main text descriptions of the coefficients proposed in Figure 4 and used in Figure 5 and 6 are difficult understand. We recommend endeavoring to create more illustrative and/or succinct descriptions.

3. Conclusions in this manuscript will be strengthened if the authors test key findings in different systems. For example, whether a let-7 3' compensatory site with +4nt offset shows a stronger repressive effect than the target with 0 or +1nt offset in cell culture; or, whether miR-155 target site with 3' paring to g15-g22 is indeed more repressed than target site with 3' paring to g12-g19 in the cells; or, measure the KD of Ago2-let-7 binding the 8mer canonical site and compare to the KD of endogenous let-7 site in lin-41. Such data would strengthen connections to findings in previous studies in the field.

4. The high affinity +4 offset let-7 data are intriguing. Are there bridging sequences that are especially good for binding? i.e. is there over-representation of specific sequences in the bridging region of the let-7 +4 binding-mode? Why is +4 offset not very often observed in nature? Why does only let-7 show two binding modes, but not miR-1 and miR-155?

*Reviewer #2 (Recommendations for the authors):*

I have only a few comments/suggestions:

a) The authors correctly note in the manuscript how some extensive pairings including position 9-10 are depleted as a result of Ago2 slicing activity, but the full impact of Ago2 endonuclease activity on the results of these experiments is not entirely clear. An informative additional experiment to address this point could have included either AGO1-miRNA complexes or a catalytically dead AGO2 mutant. A related point is that including other Argonaute proteins could uncover unexpected differences in the impact of compensatory sites on binding affinities. Finally, I wonder whether the impact of slicing activity can be inferred by sequencing the input library before and after incubation with the Ago2-miRNA complexes (of course bypassing the nitrocellulose purification step).

b) A key difference between the experimental setting used by the authors and in vivo miRNA targeting is that in cells Argonaute is post-translationally modified and acts a part of a much larger RNP complex (the RISC). It would be useful to include some simple reporter assays to test whether the different affinities for different 3'-compensatory sites observed in the AGO-RBNS experiments translates to different repressive activity in cells. For example, the authors show that a 10-20 compensatory site with a +4 offset for let-7a results in an affinity comparable to the affinity of a perfect 8-mer seed match, while the same compensatory site with -1 offset leads to only a modest increase in affinity (comparable to a 6-mer match). This particular prediction could be easily tested with appropriately-designed reporter assays.

c) On a related note, I also wonder whether some of the predictions on the efficacy of compensatory 3'-sites can be tested by looking at existing CLEAR-CLIP or CLASH datasets.

*Reviewer #3 (Recommendations for the authors):*

I found myself evaluating a paper with "invisible" results. There is a unique, technologically powerful, well-thought and most likely well-executed approach that is repeatedly applied to extract a large amount of data and derive conclusions; however, such data remains "hidden". As mentioned above, please include representative RNA bind-n-seq experiments and show how do you quantify the signals.

Unfortunately, I found the text very dense and tiresome. The data is presented in a very homogenous way and the reader has to make sense of a myriad of tiny, colored squares and other, very complex and extremely data-rich graphics. After spending (a long) time trying to understand the first two Results sections, seeing that almost the same type of figures continue until the end will tempt readers – as I was tempted – to jump to the Discussion and just get the conclusions.

I guess this is not the type of paper one feels excited to read; it is "too much". I urge the authors to re-think the format of the paper.

---

## [Author Response]

Essential revisions:All three reviewers and the reviewing editor found this to be a strong and significant manuscript that addresses important questions relevant to miRNA targeting. Nevertheless, each reviewer has made a few comments that if addressed would improve the manuscript. Specifically, it was agreed that some reporter assays as mentioned in two of the reviews would be important. Secondly, it was also agreed that the paper was quite dense and in some places difficult to read. Please consider this aspect during revision. It is important that the work be comprehensible by the general reader of eLife and not just those who appreciate all the details.

As requested, we performed reporter assays. To avoid limitations and pitfalls of standard reporter assays, we implemented a massively parallel reporter assay. With this assay, we measured repression mediated by 952 different 3′-UTR variants, which allowed us to examine with rigor the interplay between the length, position, and offset of 3′ pairing in a variety of sequence contexts. For comparison, we also measured the efficacy of sites without 3′ pairing and the efficacy of the dual sites that mediate repression of *C. elegans lin-41* mRNA. Overall, these new results, which are presented in Figure 3 of the revised manuscript, showed that key conclusions made from in vitro affinity measurements also apply to repression in cells.

To improve readability, we have condensed the portion of the paper describing experiments with chimeric and permuted miRNAs (previously Figures 5 and 6), such that only a summary of the findings is presented in the main text (current Figure 6), with the details moved into Figure 6—figure supplements 1 and 2. In addition, we have shortened the introduction and removed some of the less pertinent details throughout the results and Discussion sections.

Reviewer #1 (Recommendations for the authors):1. The study is an impressive computational feat with many interesting discoveries. However, the current manuscript is so long and time-consuming to absorb that the majority of the broader audience may lack the will to invest in reading the work and appreciating its full implications. We therefore recommend the authors shorten the manuscript with an emphasis on distilling and crystallizing the most salient points.

We have shortened the manuscript, as described above, with hopes of increasing its readability.

2. Main text descriptions of the coefficients proposed in Figure 4 and used in Figure 5 and 6 are difficult understand. We recommend endeavoring to create more illustrative and/or succinct descriptions.

We have updated the associated text with improved descriptions of the coefficients.

3. Conclusions in this manuscript will be strengthened if the authors test key findings in different systems. For example, whether a let-7 3' compensatory site with +4nt offset shows a stronger repressive effect than the target with 0 or +1nt offset in cell culture; or, whether miR-155 target site with 3' paring to g15-g22 is indeed more repressed than target site with 3' paring to g12-g19 in the cells; or, measure the KD of Ago2-let-7 binding the 8mer canonical site and compare to the KD of endogenous let-7 site in lin-41. Such data would strengthen connections to findings in previous studies in the field.

As requested, we tested some of our key findings using reporters in cell culture and included these results in our revision (Figure 3). Overall, we found that affinity in vitro corresponded well to repression in cells (*r*^2^ = 0.71). With respect to the examples mentioned by the referee, we compared the effects of the different offsets for let-7 3′-compensatory sites, and we compared the repression imparted by 8mer canonical sites to that of the endogenous let-7 sites in *lin-41*.

4. The high affinity +4 offset let-7 data are intriguing. Are there bridging sequences that are especially good for binding? i.e. is there over-representation of specific sequences in the bridging region of the let-7 +4 binding-mode? Why is +4 offset not very often observed in nature? Why does only let-7 show two binding modes, but not miR-1 and miR-155?

A and U were generally favored in the bridge region, and G was generally disfavored, which mirrored the nucleotide preferences observed flanking seed sites (McGeary, Lin, et al., 2019). Because of these preferences observed in our binding experiments, the sites for our reporter experiment were designed to have internal loop sequences formed by inserting either all-A or all-U sequences. Interestingly, 3′ sites with four U nucleotides in the bridge sequence (+4_U_-nt offset) imparted significantly less repression than those with four A nucleotides (+4_A_-nt offset) (Figure 3B). Perhaps these +4_U_ bridge sequences promoted binding to RBPs, which often bind to short oligo(U) tracts (Dominguez et al., 2018; Nostrand et al., 2020), thereby interfering with AGO–let-7a binding.

Part of the reason that the +4-nt offset is not very often observed in nature is because 3′-compensatory sites of any offset are not observed nearly as often as canonical seed-matched sites. This is presumably because of the greater information content required to acquire and maintain a high-affinity 3′-compensatory site (~6–7 nt of seed complementarity in addition to 7–9 nt of 3′ complementarity), compared to that required to acquire and maintain a high-affinity canonical site without consequential 3′ pairing (7–8 nt of seed complementarity). Likewise, among canonical sites, 3′-supplementary sites are much less common than sites with no consequential 3′ pairing, presumably because a strong 3′-supplementary site has similar information content as dual 8mer sites, yet mediates less repression than the dual sites. Thus, sites with critical 3′ pairing will be favored only in special circumstances, such as the use of a 3′-compensatory site when repression must be specific to only one member of a seed family, or the use of a 3′-supplementary site when affinity to single site must be as great as possible. The second reason that the +4 offset sites are not more common is because the optimal offset is not always +4 nt. For example, miR-124, lsy-6, and miR-7 also appear to have two different binding modes (Figure 5—figure supplement 4), but for miR-124, the optimal offset of the second binding mode is +2 nt. Indeed, our reporter-assay results suggest that even for let-7, sites with a +1-nt offset confer much greater repression than those with a 0-nt offset, which is consistent with the idea that any (small) number of positive-offset nucleotides is sufficient to enable efficient repression by 3′ sites that include pairing to more centrally located nucleotides. By these more permissive criteria, more sites can be classified as exploiting this second, positive-offset binding mode, including the let-7 sites in *lin-14*, one of the two lsy-6 sites in *cog1*, and the miR-7 site in Cyrano.

We propose that let-7a preferentially benefits from the positive-offset binding mode because it has a GG dinucleotide at positions 11 and 12, and the positive-offset binding mode provides the most favorable access for pairing to this high-affinity G11-G12 dinucleotide. Supporting this proposal, re-analyses of results from random-sequence libraries (McGeary, Lin, et al., 2019) indicate that miR-124, lsy-6, and miR-7 each show two different binding modes (Figure 5—figure supplement 4), which we attribute to the G11-G12 dinucleotide of miR-124, and to the G11 nucleotide of both lsy-6 and miR-7.

Reviewer #2 (Recommendations for the authors):I have only a few comments/suggestions:a) The authors correctly note in the manuscript how some extensive pairings including position 9-10 are depleted as a result of Ago2 slicing activity, but the full impact of Ago2 endonuclease activity on the results of these experiments is not entirely clear. An informative additional experiment to address this point could have included either AGO1-miRNA complexes or a catalytically dead AGO2 mutant. A related point is that including other Argonaute proteins could uncover unexpected differences in the impact of compensatory sites on binding affinities. Finally, I wonder whether the impact of slicing activity can be inferred by sequencing the input library before and after incubation with the Ago2-miRNA complexes (of course bypassing the nitrocellulose purification step).

The proposed additional experiments would indeed be helpful for better understanding the quantitative contribution of slicing to the depletion observed for 3′-compensatory sites with contiguous central pairing (Figure 2C). However, because these sites constitute only a small minority of the analyzed sites, and because our paper is already quite lengthy, we have chosen not to pursue these experiments for the current manuscript. Instead, we have revised our manuscript to make it clearer that our evidence of slicing is indirect. With respect to other Argonaute proteins, it would indeed be interesting if they might impart different 3′-binding preferences, but again because our paper is already quite lengthy, we have chosen not to pursue these experiments for the current manuscript.

b) A key difference between the experimental setting used by the authors and in vivo miRNA targeting is that in cells Argonaute is post-translationally modified and acts a part of a much larger RNP complex (the RISC). It would be useful to include some simple reporter assays to test whether the different affinities for different 3'-compensatory sites observed in the AGO-RBNS experiments translates to different repressive activity in cells. For example, the authors show that a 10-20 compensatory site with a +4 offset for let-7a results in an affinity comparable to the affinity of a perfect 8-mer seed match, while the same compensatory site with -1 offset leads to only a modest increase in affinity (comparable to a 6-mer match). This particular prediction could be easily tested with appropriately-designed reporter assays.

As requested, we performed reporter assays for let-7a sites and included these results in our revision (Figure 3). Overall, we found that affinity in vitro corresponded well to repression in cells (*r*^2^ = 0.71). Regarding the example mentioned by the referee, we found that a compensatory site with complementarity to positions 11–19 and a positive offset typically was as efficacious as a perfect 8mer seed match, whereas the same compensatory site with a 0-nt offset was substantially less efficacious.

c) On a related note, I also wonder whether some of the predictions on the efficacy of compensatory 3'-sites can be tested by looking at existing CLEAR-CLIP or CLASH datasets.

Although CLIP/CLASH data do identify sites of endogenously engaged miRNAs, quantitative analysis can be confounded by the differential crosslinking efficiencies observed in different sequence contexts. Thus, we opted instead to use a massively parallel reporter assay to test predictions on the efficacy of compensatory 3′ sites.

Reviewer #3 (Recommendations for the authors):I found myself evaluating a paper with "invisible" results. There is a unique, technologically powerful, well-thought and most likely well-executed approach that is repeatedly applied to extract a large amount of data and derive conclusions; however, such data remains "hidden". As mentioned above, please include representative RNA bind-n-seq experiments and show how do you quantify the signals.

First of all, we appreciate this sincere desire to help us improve our paper. The difficultly stems from the fact that our primary data are the sequences of millions of reads, not labeled molecules on a nitrocellulose filter, and thus these data are a step removed from what we can show in a representative experiment. We considered illustrating in more detail how the relative *K*_D_ values are extracted using the counts from all of the binding samples, as we had done in Figure 1C–E of McGeary, Lin, et al., (2019). However, we were concerned that this would bog down the beginning of the manuscript. We now cite these figure panels within the Figure 2A legend, so that interested readers can more easily access visuals that might help them better understand the computational workflow.

Unfortunately, I found the text very dense and tiresome. The data is presented in a very homogenous way and the reader has to make sense of a myriad of tiny, colored squares and other, very complex and extremely data-rich graphics. After spending (a long) time trying to understand the first two Results sections, seeing that almost the same type of figures continue until the end will tempt readers – as I was tempted – to jump to the Discussion and just get the conclusions.I guess this is not the type of paper one feels excited to read; it is "too much". I urge the authors to re-think the format of the paper.

We greatly appreciate the effort expended to understand our results, and recognize that a typical reader will not have this level of commitment. We have attempted to condense the narrative and remove less pertinent details, moving the results from the experiments using chimeric and permuted miRNAs into figure supplements. We also hope that the added reporter experiments, which are somewhat more conventional, will be a welcome interlude from the RBNS analyses and help make the manuscript more readable.